# A surface strategy boosting the ethylene selectivity for CO$_2$ reduction and in situ mechanistic insights

Yinchao Yao[1,6], Tong Shi[2,3,6], Wenxing Chen [1], Jiehua Wu[4], Yunying Fan[5], Yichun Liu[5], Liang Cao [2] ✉ & Zhuo Chen [1] ✉

Electrochemical reduction of carbon dioxide into ethylene, as opposed to traditional industrial methods, represents a more environmentally friendly and promising technical approach. However, achieving high activity of ethylene remains a huge challenge due to the numerous possible reaction pathways. Here, we construct a hierarchical nanoelectrode composed of CuO treated with dodecanethiol to achieve elevated ethylene activity with a Faradaic efficiency reaching 79.5%. Through on in situ investigations, it is observed that dodecanethiol modification not only facilitates CO$_2$ transfer and enhances *CO coverage on the catalyst surfaces, but also stabilizes Cu(100) facet. Density functional theory calculations of activation energy barriers of the asymmetrical C–C coupling between *CO and *CHO further support that the greatly increased selectivity of ethylene is attributed to the thiol-stabilized Cu(100). Our findings not only provide an effective strategy to design and construct Cu-based catalysts for highly selective CO$_2$ to ethylene, but also offer deep insights into the mechanism of CO$_2$ to ethylene.

The electrocatalytic conversion of carbon dioxide into high-value-added chemicals and fuels stands as a promising technological avenue for transforming waste into valuable resources and fostering the development of a sustainable, carbon-neutral society[1]. Among various carbon dioxide reduction products, C$_{2+}$ products are attracting attention due to their higher value, especially for ethylene as an essential building block in the chemical industry, emerging as a focal point in this realm[2–7]. However, the efficient conversion of CO$_2$ to ethylene remains a formidable challenge. Notably, copper-based catalysts within the realm of transition metal catalysts have attracted considerable interest due to their excellent C$_{2+}$ product selectivity[3,5,6,8]. In general, the electrocatalytic CO$_2$ reduction reaction (CO$_2$RR) over copper-based catalysts involves three pivotal steps: efficient CO$_2$ diffusion into the catalyst interface, activation of CO$_2$ molecules at the interface, and concerted proton–electron transfer (CPET) steps, culminating in the formation of *CO intermediates[9,10]. Lastly, catalysts must exhibit high activity in promoting the C–C coupling reaction to facilitate the production of C$_{2+}$ products.

With considerable attention devoted to overcoming the limitations of the two formers, some effective strategies have been developed, such as developing efficient electrocatalysts[11], tailoring electrode–electrolyte interface[12], and optimizing the electrolyser design[4]. However, the C–C coupling is the most critical step in determining the selectivity and yield of various C$_{2+}$ products[13–15]. Moreover, there persists a lack of comprehensive consensus on the C–C coupling mechanism[10,14]. Numerous approaches have been reported to promote

[1]Energy & Catalysis Center, Department of Materials Physics and Chemistry, School of Materials Science and Engineering, Beijing Institute of Technology, 100081 Beijing, PR China. [2]Institute of Catalysis, Department of Chemistry, Zhejiang University, Hangzhou 310058 Zhejiang, PR China. [3]Inner Mongolia Key Laboratory of Chemistry and Physics of Rare Earth Materials, Department of Chemistry and Chemical Engineering, Inner Mongolia University, Hohhot 010021, PR China. [4]SINOPEC (Beijing) Research Institute of Chemical Industry Co., Ltd, 100013 Beijing, PR China. [5]School of Materials Science and Engineering, Kunming University of Science and Technology, Kunming 650093, PR China. [6]These authors contributed equally: Yinchao Yao, Tong Shi. ✉e-mail: liangcao@zju.edu.cn; zchen@bit.edu.cn

the C–C coupling on copper surfaces, including morphology tuning[9,16], crystal facet regulation[17,18], chemical state manipulation[7,19,20], and surface modification[3,21], with crystal facet being identified as a crucial parameter. According to the crystal structure of copper, Cu(100) with lower surface coordination numbers exhibits higher CO dimerization reactivity and $C_{2+}$ product selectivity[18,22,23]. Theoretical calculations indicate thermodynamic favorability for C–C coupling of the two *CO intermediates on Cu(100) to produce ethylene[15,24,25]. Thus, controlled exposure of Cu(100) is deemed essential for facilitating ethylene selectivity. Experimental studies confirm that Cu cubes with rich (100) facets significantly increase the ethylene Faraday efficiency to 57%[26]. However, dynamic reconstruction of the copper crystal face during $CO_2RR$, driven by its low cohesion energy, poses challenges in maintaining the crystal structure, exacerbated by $CO_2RR$ intermediates[22,27]. Therefore, to further enhance the selectivity of ethylene, it is imperative to improve the Cu(100) content of the catalyst while ensuring the stability of the Cu(100) crystalline surface during the $CO_2RR$ reaction.

Herein, we employ a DDT-functionalized CuO hierarchical nanostructural electrode to achieve high selectivity of $CO_2RR$ toward ethylene. The Faradaic efficiency (FE) of $C_2H_4$ product reaches up to 72%, marking an enhancement of more than 4-fold compared to the electrode without DDT treatment. In situ Raman and attenuated total reflection-Fourier transform infrared (ATR-FTIR) spectroscopy results indicate that DDT facilitates $CO_2$ transport and enhances CO coverage on the catalyst surface. Moreover, in situ XRD and X-ray absorption spectroscopies (XAS) investigations elucidate that DDT stabilizes Cu(100) facets, thus promoting the C–C coupling. DFT[28] calculations further reveal that thiol stabilized Cu(100) facet can reduce the activation energy barrier of C–C coupling between *CO and *CHO, in which the intermediates are confirmed by in situ ATR-FTIR. This work paves a promising route for highly selective $CO_2RR$ to ethylene through surface-modified Cu-based catalysts and deepens insights into the mechanism of $CO_2RR$ to ethylene.

## Results and discussion
### Catalyst design and preparation
The electrocatalytic $CO_2RR$ occurring at the interface between the electrode and electrolyte encompasses multiple stages: the diffusion of $CO_2$ onto the electrode surface, subsequent adsorption at the reaction site, concurrent transfer of electrons and protons, reduction to hydrocarbons, and eventual desorption. To address the simultaneous the requirements of this aspects, we engineered a hierarchical nano-structural electrode with a hydrophobic surface as shown in Fig. 1a. Specifically, CuO nanowire arrays were prepared on a copper foam substrate through annealing the electrodeposited $Cu(OH)_2$ nanowire arrays, leading to a hierarchical nanostructure, and thus not only facilitating charge transfer via the ordered 1D channel but also exposing more active sites through the high surface area. Scanning electron microscopy (SEM) and transmission electron microscopy (TEM) confirmed the orderly arranged nanowire morphology of CuO (Supplementary Fig. 1). High-resolution TEM (HRTEM) and X-ray diffraction pattern showed that the crystal structure of nanowires was composed of $Cu_2O$ and CuO (Supplementary Figs. 1 and 2). To build a hydrophobic interface while concurrently stabilizing the Cu(100) surface, we conducted a screening of potential molecules via DFT calculations combined with literature research. Based on DFT calculations of the adsorption energies of various alcohols, thiols, and amines with alkyl chains (inset in Fig. 1a), we selected the thiol molecule with the highest binding ability to the Cu(100) facets for surface modification. Additionally, the calculations indicated a stronger binding of the thiol molecule to Cu(100) than Cu(111), suggesting a preference for stabilizing the Cu(100) surface, as shown in Fig. 1a, Supplementary Fig. 3 and Supplementary Table 1, meaning that the thiol molecule can preferentially stabilize the Cu(100) surface. Consequently, a hydrophobic thiol salt layer was introduced on the CuO surface by immersing the

CuO nanoarrays into liquid DDT, creating a three-phase interface and thereby enhancing local $CO_2$ concentration. For convenience, the electrodes before and after DDT modification are denoted by CuO and CuO-SH, respectively. The contact angle test results show that a superhydrophobic interface was formed on the CuO-SH electrode's surface (Fig. 1b). Expectedly, the hierarchical nanostructure of CuO-SH was confirmed by SEM and TEM (Fig. 1b, c). Energy-dispersive X-ray spectroscopy (EDS) analyses in a STEM showed a uniform distribution of S and C environments on the CuO-SH outer layer (Fig. 1d), confirming the presence of DDT. The functionalized catalyst showed a notable shift in the absorption edge observed in X-ray absorption near-edge structure (XANES) spectra (Fig. 1e) compared to pristine CuO, from 8981.65 eV to 8979.73 eV in First derivative normalized absorption (inset in Fig. 1e)[29,30], indicating the DDT reduced $Cu^{2+}$ species to form a surface of $Cu^{1+}$ (Fig. 1f). A new S2p peak was detected at 163 eV consistent with the Cu-S bonds (Fig. 1g)[31]. The successful functionalization with the DDT was further corroborated through attenuated total reflection-Fourier transform infrared (ATR-FTIR) spectroscopy (Supplementary Fig. 4).

### CO₂RR catalytic performance
To evaluate the $CO_2RR$ activity of the DDT-modified hierarchical nanoelectrode, the electrocatalytic $CO_2RR$ performance was measured in an H-type electrochemical cell with a $CO_2$-saturated 0.1 M $KHCO_3$ electrolyte at applied potentials from −0.8 to −1.6 V. The products were characterized by nuclear magnetic resonance (NMR) and gas chromatography (GC) (See details in the Methods section), including CO, formate, $CH_4$, and $C_{2+}$ products (ethylene and ethanol). Figure 2a illustrates ethylene as the primary product of the CuO-SH catalyst, with ethylene selectivity gradually increasing with rising overpotential, reaching a peak Faraday efficiency (FE) of 72% at −1.4 V. In contrast, the unmodified CuO electrode shows a diverse distribution of products, comprising $C_1$ products (CO, formate) and $H_2$ (Fig. 2b). As the applied potential decreases, the FE of $C_{2+}$ products for CuO diminishes, and the FE for $H_2$ rises to 58.9% at −1.6 V, where $H_2$ becomes the predominant product. However, the FE of $H_2$ for CuO-SH at the same potential is less than 10%, indicating a substantially suppressed HER. To visually represent the influence of DDT modification on the $C_{2+}$ products, the data is re-plotted in Fig. 2c. At −1.6 V, the $C_{2+}$ FE of the CuO-SH electrode is 79.6%, contrasting with 16.9% for the original CuO electrode, with ethylene comprising the majority of $C_{2+}$ products, showcasing an enhancement of more than 4-fold. The $C_{2+}/C_1$ ratio in the $CO_2RR$ products was also calculated. The ratio of 7.8 for CuO-SH at −1.4 V indicates a preference for CO intermediates to dimerize, producing $C_{2+}$ products rather than $C_1$ species. In comparison, the $C_{2+}/C_1$ products ratio on CuO (0.82 at −1.4 V) is much lower than that on CuO-SH. Furthermore, the ratio of FE for $C_2H_4$ products to hydrogen ($FEC_2H_4/FEH_2$) (Supplementary Fig. 5) further verifies the boosted activity of the C–C coupling steps on CuO-SH electrodes.

To illustrate the potential of $CO_2$ electrolysis for large-scale ethylene production, we employed a gas-diffusion-electrode-based flow-cell system equipped with the CuO-SH catalyst. The FE for ethylene product formation on the CuO-SH electrode is further augmented, reaching a maximum FE of 79.5% at −1.2 V, accompanied by a total current density of 304 mA cm⁻². Notably, the ethylene partial current density achieved an impressive 242 mA cm⁻², surpassing the majority of previous reports (Fig. 2d and Supplementary Table 2). In contrast, the highest $C_2H_4$ FE and partial current density on the CuO catalyst were 44.8% and 169.3 mA cm⁻² (Fig. 2e). Furthermore, a prolonged stability test of CuO-SH in a flow cell was conducted. The ethylene FE of CuO-SH initiated at 73.6% and remained consistently high at 49.7% after 40 h of testing under a constant current density of 200 mA cm⁻², demonstrating superior stability compared to CuO, which dropped to less than 10% over the same duration (Supplementary Fig. 6). Consequently, DDT treatment not only significantly

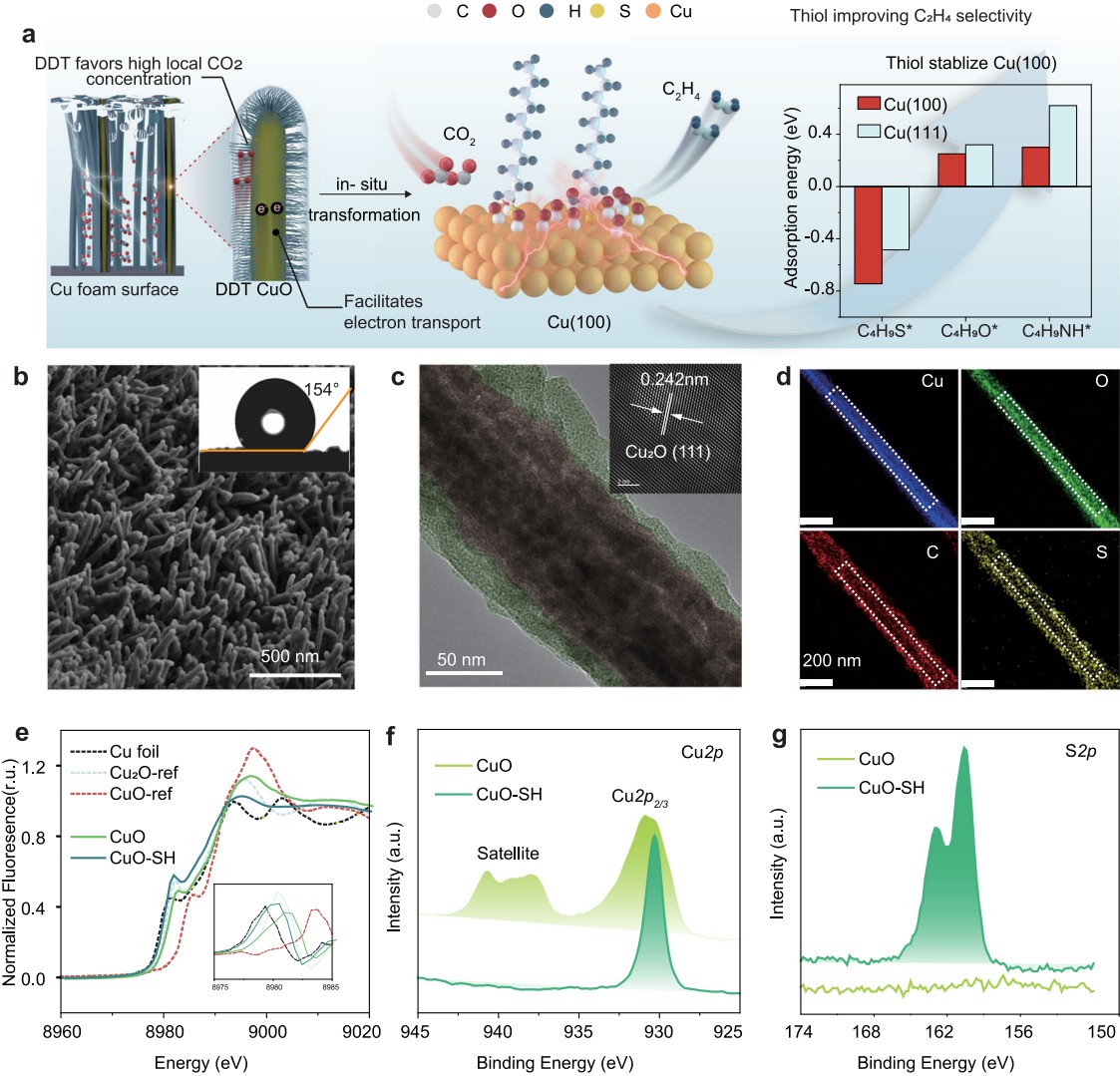

**Fig. 1 | Schematic diagram of design and preparation process and characterizations of the DDT functionalized CuO-SH catalysts. a** Schematic illustration for the design and preparation process of DDT modification strategy, and (Inset) DFT calculations of the adsorption energies of various alcohols, thiols, and amines with alkyl chains. **b** Scanning electron microscope (SEM) image of CuO-SH electrode, (Inset) the contact angle measurements of CuO-SH. **c** TEM image, (Inset) FFT of HRTEM image and (**d**) EDS elemental mapping images of CuO-SH. **e** XANES spectra (**f**) Cu2*p*, (**g**) S2*p* XPS curves for CuO-SH and CuO.

enhances ethylene selectivity but also improves stability, showcasing its potential for practical applications in large-scale ethylene production through $CO_2$ electrolysis.

Despite the achieved increase in the FE of ethylene on CuO-SH through DDT modification, a gradual decrease in ethylene FE was observed during the stability test (Supplementary Figs. 6 and 7). Subsequently, an investigation was conducted to assess the stabilities of the thiol molecule on the CuO-SH electrode during the electrocatalytic process. The S2*p*, S *K*-edge XAFS and STEM-EDS mapping results confirm that the majority of DDT molecules remain during the $CO_2$RR (Supplementary Figs. 8 and 9). However, a new peak attributed to alkanesulfonates[7] appeared at 168 eV in the S2*p* spectrum of CuO-SH after 5 h of reaction, becoming more pronounced with an extended reaction time of 40 h. Additionally, the S *K*-edge XAFS of the CuO-SH catalyst revealed an increase in the surface S valence state with prolonged reaction time at −1.4 V (Supplementary Fig. 8b). Further analysis of the S2*p* spectrum of CuO-SH revealed that, although most of the DDT remained intact, there was still a 17.3% DDT loss which was confirmed by the repetitive experiments (Supplementary Fig. 10). Consequently, we attribute the performance degradation of CuO-SH primarily to the partial loss of thiol groups, either through oxidation or

detachment (Supplementary Fig. 11). To further verify the impact of thiols on stability, thiols were used to retreat the CuO-SH samples after stability testing. The retreatment achieved a 27.5% increase in surface thiols recovery, leading to an improvement in the FE of ethylene (from 46.3 to 64.1%) (Supplementary Fig. 12). Following two rounds of retreatment, the ethylene FE consistently maintained a relatively high level (52.1%) after 160 h of operation, demonstrating the enhanced stability of the CuO-SH catalyst (Fig. 2f).

We infer that the improvement in $C_2H_4$ selectivity resulting from DDT modification can be attributed to three key contributions: enhanced $CO_2$ transport, increased coverages of *CO, and intrinsic activity. As mentioned earlier, DDT induces a hydrophobic surface on the electrode, leading to the formation of a three-phase interface. The ATR-FTIR spectrum indeed reveals that this hydrophobic interface enhances the local $CO_2$ concentration, indicating that DDT facilitates $CO_2$ transportation and inhibits the competing HER reaction. (Supplementary Figs. 13 and 14).

To clarify the effect of DDT on the *CO coverages during $CO_2$RR, we conducted a potential-dependent in situ Raman spectroscopy study in a $CO_2$-saturated 0.1 M KHCO$_3$ electrolyte, enabling real-time monitoring surface adsorbates and quantification of adsorbed *CO

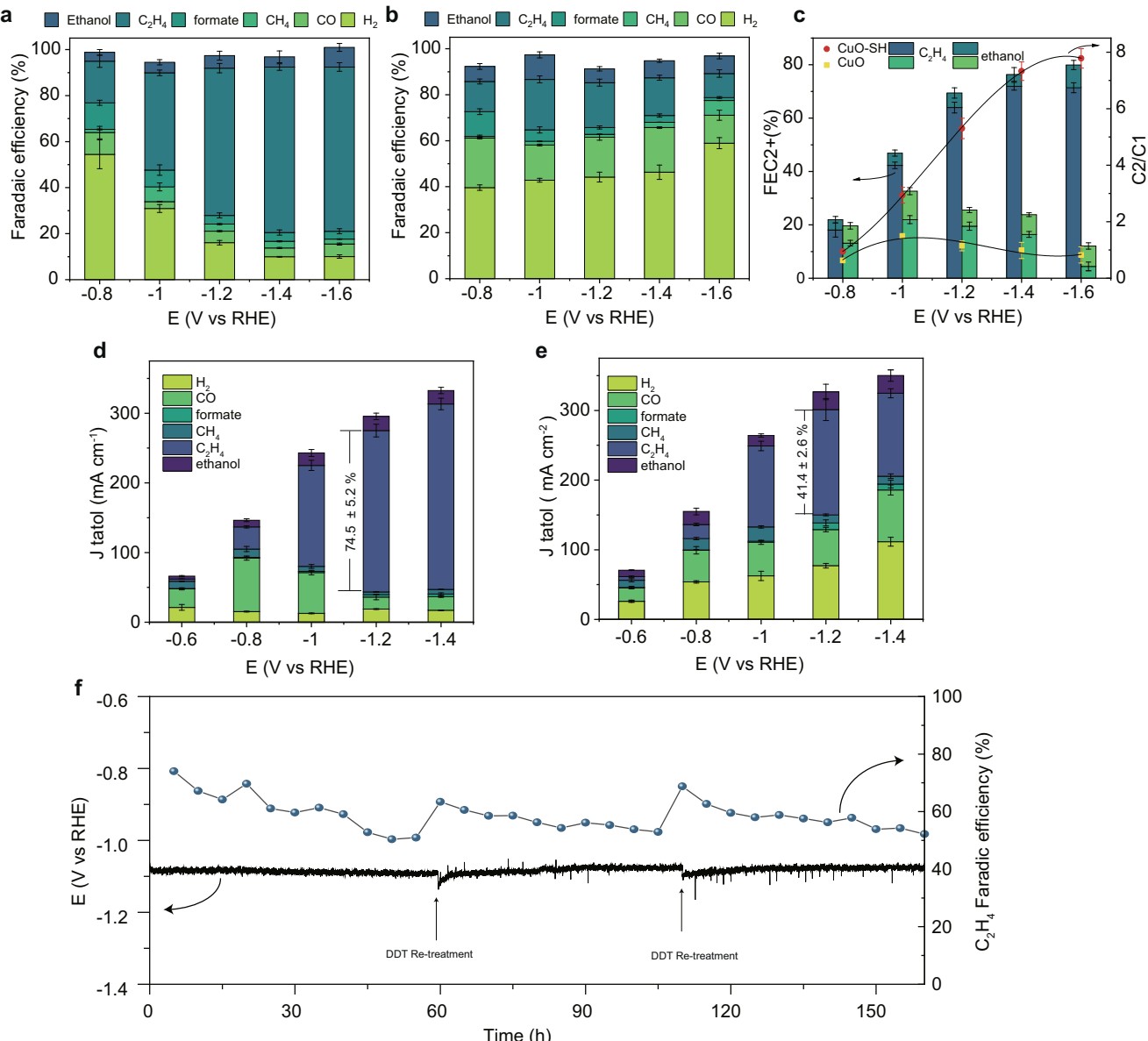

**Fig. 2 | CO₂RR performances for the CuO and CuO-SH.** Faradaic efficiencies (FE) of the CO₂RR products as a function of applied potential over (**a**) CuO-SH and (**b**) CuO, **c** FE values for C₂₊ products and FE ratio of C₂₊ products over C₁ products on CuO-SH and CuO at various potentials ranging from −0.8 to −1.6 V, current densities and product distributions at different potentials over (**d**) CuO-SH and (**e**) CuO under flow cell measurement, **f** Stability test for CuO-SH at the current density 200 mA cm⁻² in a flow cell. The potential was corrected with 85% iR compensation. Error bars indicate the standard deviation of three independent measurements.

coverages at a range of applied potentials from the open-circuit potential (OCP) to −1.2 V on CuO-SH and CuO catalysts. Figure 3a, b illustrates that both electrodes exhibit two peaks at -1313 and -1616 cm⁻¹, attributed to the glassy carbon substrate[32]. An absorption band centered at -2057 cm⁻¹ starts to appear from a cathodic potential of −0.2 V for both CuO-SH and CuO catalysts, typically assigned to C≡O stretching of the atop-adsorbed *CO species[33]. The peak intensity of CO adsorption gradually increases when shifting to a more negative potential, indicating a higher *CO coverage derived from the activated CO₂. For a more accurate comparison of the CO peak intensities between the two samples, we normalize the CO characteristic peaks using the glassy carbon peaks as a reference (Supplementary Fig. 15). As depicted in Fig. 3c, the CuO-SH catalyst exhibits a higher CO peak intensity than CuO, signifying that DDT surface modification enhances CO absorption strength. Simultaneously, it is observed that the *CO atop Raman peak shifted toward lower Raman shift (Fig. 3d). By comparing the peak positions of the CO adsorption peaks, we note

that within this potential range, the CO adsorption peak of CuO-SH is slightly lower than that of CuO, indicating a stronger CO binding ability of CuO-SH.

In addition to CO₂ transport and the adsorption of *CO intermediates, the C-C coupling is paramount for achieving high C₂H₄ selectivity. We further explored the impact of DDT modification on the intrinsic activity of the catalyst for the C-C coupling. To minimize the effect of the electrochemically active surface area (ECSA), we employed the double layer capacity (Cdl) method to measure the ECSA of the electrodes. The CuO-SH electrode exhibited an ECSA of 11.9 mF cm⁻², lower than that of unmodified CuO (16.4 mF cm⁻²) (Supplementary Fig. 17c). Subsequently, we normalized the partial current density of C₂H₄ products based on ECSA (Supplementary Fig. 17d). Despite the reduction in ECSA due to DDT modification, the CuO-SH delivers a superior ECSA-normalized C₂H₄ current density (2.93 mA mF⁻¹) compared to CuO (0.12 mA mF⁻¹) at −1.6 V, indicating that DDT significantly enhances the intrinsic activity of the C-C coupling.

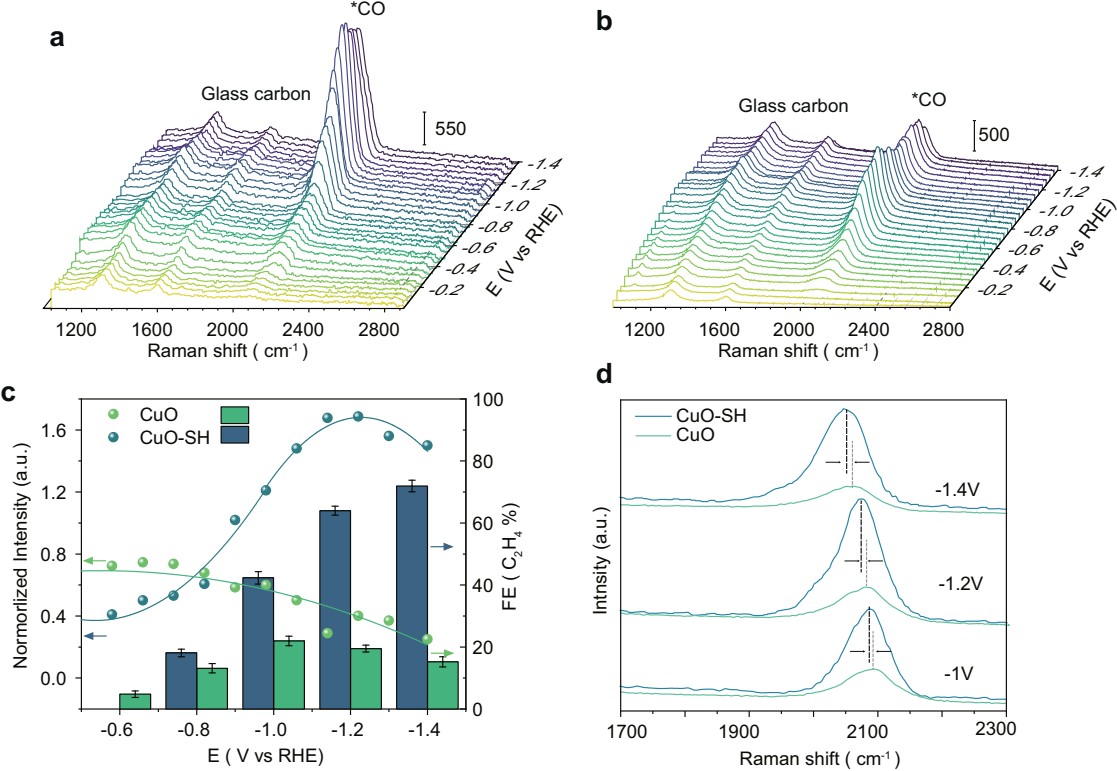

**Fig. 3 | In situ Raman spectra.** In situ Raman spectra of (**a**) CuO-SH and (**b**) CuO obtained in a potential window OCP to −1.4 V, **c** Potential dependence of the normalized intensity of *CO and FE of $C_2H_4$ (error bars indicate the standard deviation of three independent measurements), **d** in situ Raman spectra of CuO-SH and CuO collected at different applied potential.

To gain deeper insights into the improved intrinsic activity of the C-C coupling, in situ XAS experiments were conducted to discern the evolution of the local structure of the catalysts. XANES spectra showcased a shift in absorption edge toward lower energies, indicating the reduction of cupric oxides during $CO_2RR$. Both the CuO-SH catalysts and the unmodified CuO samples underwent conversion to metallic Cu, as evidenced in Supplementary Fig. 18. Subsequently, we conducted fitting of the extended X-ray absorption fine structure (EXAFS) spectra at the Cu *K*-edge of both CuO-SH and CuO samples (Fig. 4a, b and Supplementary Figs. 19 and 20). The Cu–Cu coordination number in CuO-SH is around 9 at −1.6 V (Fig. 4c and Supplementary Table 3), which is lower than that of bulk face-centered cubic Cu and CuO, while that of the pristine CuO exceeded 10. This implies the formation of more unsaturated coordination structures in CuO-SH, serving as potential active sites.

We then conducted operando XRD experiments to scrutinize the structural transitions of CuO-SH and CuO catalysts during $CO_2RR$. As depicted in Fig. 4d, e, the characteristic peaks of copper oxide gradually diminished and disappeared at −0.8 V. In contrast, the intensity of XRD peaks corresponding to metallic Cu(111) and Cu(100) increased as the potential gradually shifted to −1.4 V, as confirmed by HRTEM imaging (Supplementary Fig. 21). According to surface energy calculations[18], the Cu(111) facet with a surface energy of 1.25 J cm$^{-2}$ is more stable than Cu(100) with a surface energy of 1.43 J cm$^{-2}$. Interestingly, the XRD results of Cu nanocatalysts derived from CuO-SH exhibited a higher intensity of the Cu(100) peaks compared to the CuO sample. Furthermore, we plotted the potential-dependent (100)/(111) crystal plane ratio and $C_2H_4$ selectivity trend to elucidate the contribution of crystal structure to ethylene selectivity (Fig. 4f). After normalization of XRD peak intensities, the Cu(100)/(111) intensities ratio of CuO-SH was over 3 times that of CuO at −1.4 V. Given that different copper facets display distinct OH⁻ electrochemical adsorption behavior, the OH⁻ electroabsorption experiment was conducted to

further analyze the surface structure of CuO and CuO-SH at the potentials ranging from −1 to −1.4 V. As shown in Supplementary Fig. 22, linear sweep voltammetry profiles revealed electrochemical OH⁻ adsorption peaks on Cu(100), Cu(110) and Cu(111) at potentials 0.36, 0.42, and 0.47 V versus RHE, respectively. In the OH⁻ absorption analysis, we observed a high Cu(100)/Cu(111) ratio of 1.16, 1.04, and 1.09 at potentials ranging from −1 to −1.4 V vs. RHE, respectively, consistent with the results of the in situ XRD measurements. This trend indicates that DDT molecules stabilize Cu(100), leading to the preferential generation of Cu(100) surfaces during $CO_2RR$. Additionally, the in situ XRD of DDT-treated Cu(100) single crystal foil further confirms that DDT is beneficial for stabilizing Cu(100) during the $CO_2RR$ reaction (Supplementary Fig. 23).

A positive correlation exists between the FE of ethylene and the (100)/(111) ratio of Cu derived from CuO and CuO-SH (Fig. 4f). To elucidate which crystal facet exhibits higher activity for ethylene selectivity, we employed well-defined Cu(100) and Cu(111) single crystal foil as a simplified model to eliminate the influence of other factors. The corresponding $CO_2RR$ performances indicate that although the ethylene conversion efficiency of both single crystal foils is not high, Cu(100) unequivocally demonstrates a higher ethylene FE than Cu(111) (Supplementary Figs. 24 and 25). To understand the impact of catalyst facets on ethylene selectivity under real condition, we further investigated the time-dependent ethylene selectivity with the structural evolution of CuO-SH. We collected the time-resolved in situ XRD of CuO-SH at −1.4 V in 0.1 M $KHCO_3$ and also conducted time-dependent performance testing over the CuO-SH electrode at −1.4 V (Supplementary Fig. 26). To observe the transformation process of crystal planes, it is worth noting that the CuO-SH samples used in this test were not subjected to activation treatment. As shown in Supplementary Fig. 26, the diffraction peak of Cu(100) gradually increases within the first 40 min, and the ratio of Cu(100)/Cu(111) stabilizes around 1 after 60 min. Correspondingly, the ethylene FE also

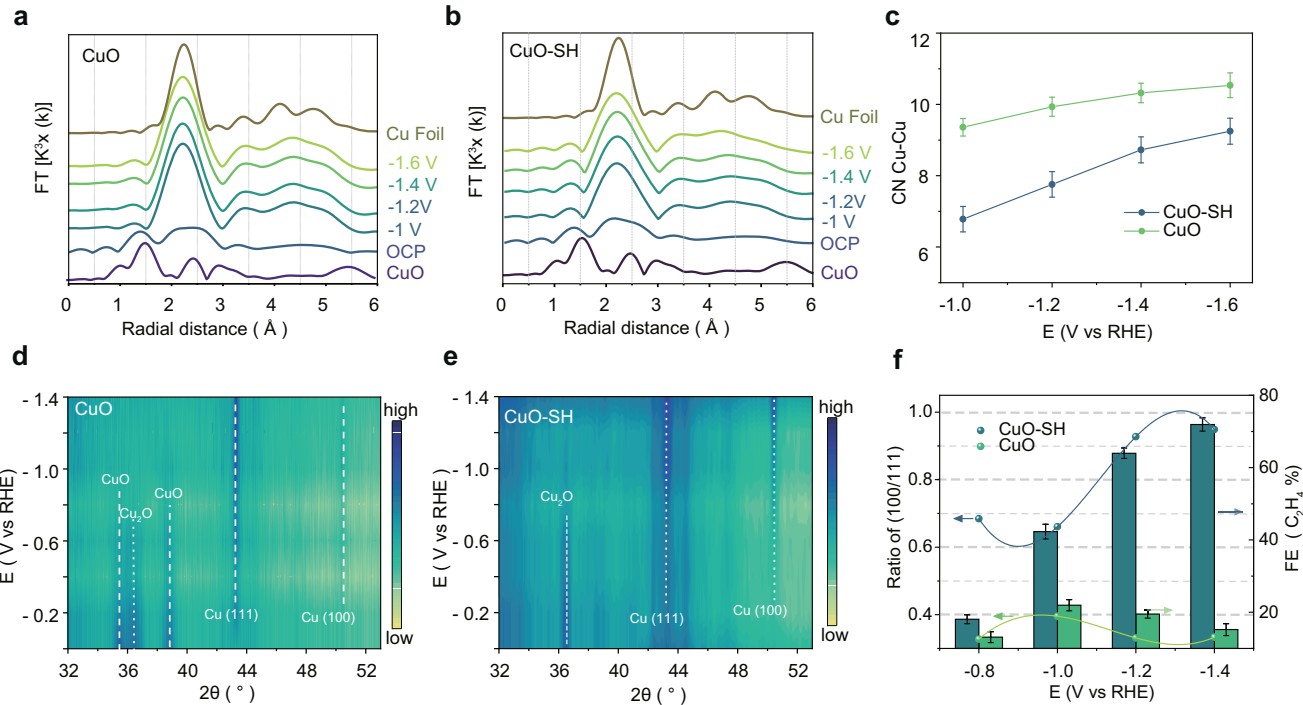

**Fig. 4 | In situ XAFS and in situ XRD results.** Fourier transform curves of in situ EXAFS at the Cu *K*-edge of the (**a**) CuO-SH and (**b**) CuO obtained in a potential window −1 to −1.6 V, **c** Coordination numbers of the first intermetallic Cu-Cu shell (error bars represent the fit error), In situ XRD patterns of (**d**) CuO-SH and (**e**) CuO collected at various potentials ranging from 0 to −1.4 V, **f** Quantitative peak analysis: the ratio of Cu(100) and Cu(111) facets (error bars indicate the standard deviation of three independent measurements).

gradually increases with Cu(100) formation within the first 35 min and then reaches the maximum value. Whether using single crystal Cu foils or examining the time-dependent ethylene selectivity with structural evolution, it has been reaffirmed that Cu(100) possesses a higher ethylene FE than Cu(111), and the improved Cu(100) is responsible for the enhancement of ethylene selectivity in CuO-SH.

We probed the structural stability of the CuO-SH catalyst through the stability test. The time-dependent in situ XRD at −1.4 V showed that the CuO-SH catalyst then maintains a relatively stable crystal structure after being reduced (Supplementary Fig. 26b). Additionally, the XRD results of CuO-SH after the 40 h stability test indicated a slight reduction in the Cu(100) facets of the catalyst, as evidenced by a decrease in the Cu(100)/Cu(111) ratio from 1.09 at 2 h to 0.87 at 40 h (Supplementary Fig. 27). The reduction of Cu(100) facets may be attributed to the partial conversion of DDT, thus leading to a decrease in the ethylene FE of the CuO-SH in the stability tests.

### Mechanistic studies

As proposed in prior literature, the generation of multi-carbon products during $CO_2RR$ could occur through the dimerization of two *CO intermediates[24,34] or *CO-COH intermediates[35–37]. Therefore, the ATR-FTIR was performed to elucidate the C−C coupling mechanism on CuO-SH during $CO_2RR$. The IR spectra were collected from the OCP to −1.4 V after the pre-reduction of both the CuO and CuO-SH catalysts. The peaks at -2043 cm$^{-1}$ and -1890 cm$^{-1}$ correspond to the atop-adsorbed *CO ($CO_L$) and bridge-bond CO ($CO_B$) intermediates, respectively, which are widely used as indicators of the CO coverage over the catalysts surfaces[34,38–40]. Moreover, a marked tuning of *$CO_L$ is observed as the potential becomes more negative (Supplementary Fig. 29 and Supplementary Table 4). The CuO-SH catalysts show a stronger infrared band for *$CO_L$ compared to the unmodified CuO (Fig. 5a, b and Supplementary Fig. 30). The peak position of CO adsorbed on CuO-SH closely overlapped with that on Cu(100) foil, while the CO peak position of CuO is closer to Cu(111) foil. Based on the

in situ IR spectra through an internal standard, the Cu(100) foil showed a greater normalized CO intensity, indicating stronger CO absorption compared to the Cu(111) foil, in agreement with our DFT results (Supplementary Figs. 31–33). The improved coverage of *$CO_L$ could be attributed to the increased proportion of Cu(100) facets on the CuO-SH surface. These variations in CO absorption behavior indicate that the DDT treatment improves the binding strength of *CO on CuO-SH, leading to enhanced *CO coverage, in agreement with the Raman experiments. Notably, the *CHO species adsorbed on CuO-SH are observed in Fig. 5a, with peak[41] located at -1720 cm$^{-1}$, representing a crucial intermediate for C−C asymmetric coupling.

Our DFT results indicated that the *CHO intermediate on CuO-SH derived from *$CO_L$ has a lower protonation energy barrier on Cu(100) than *$CO_B$ (Supplementary Fig. 34). Furthermore, the additional peak at -1586 cm$^{-1}$ and -1236 cm$^{-1}$ collected on the CuO-SH catalyst are indexed to the adsorbed *CO-CHO intermediates (Fig. 5a)[42]. Therefore, the C−C coupling on CuO-SH could be asymmetrically triggered between *CO and *CHO. The ATR−FTIR results provide experimental evidence of the asymmetric coupling process.

Finally, we performed DFT calculations to gain a deeper understanding the mechanism underlying the significantly enhanced activity and selectivity toward the ethylene due to DDT modification. To truncate the carbon chain length of thiol used for DFT calculations, the adsorption energies of thiols with varied carbon chain lengths were calculated. *$C_4H_9S$ was selected because the adsorption energy converges on the number of carbon atoms reaching four (Supplementary Fig. 35). *CO-*CHO was experimentally characterized as the prominently intermediate adsorbate on thiol-modified copper surfaces (Fig. 5a), indicating the necessity of investigating the activation energy barrier of CO-CHO coupling. Due to the sluggish kinetics and the high thermodynamic energy barrier of the C-C coupling step, it is commonly considered the rate-limiting step for $CO_2RR$[43]. The activation free energy barriers of CO-CHO coupling were calculated on four representative surfaces of Cu(100), Cu(100) with *$C_4H_9S$, Cu(111) and

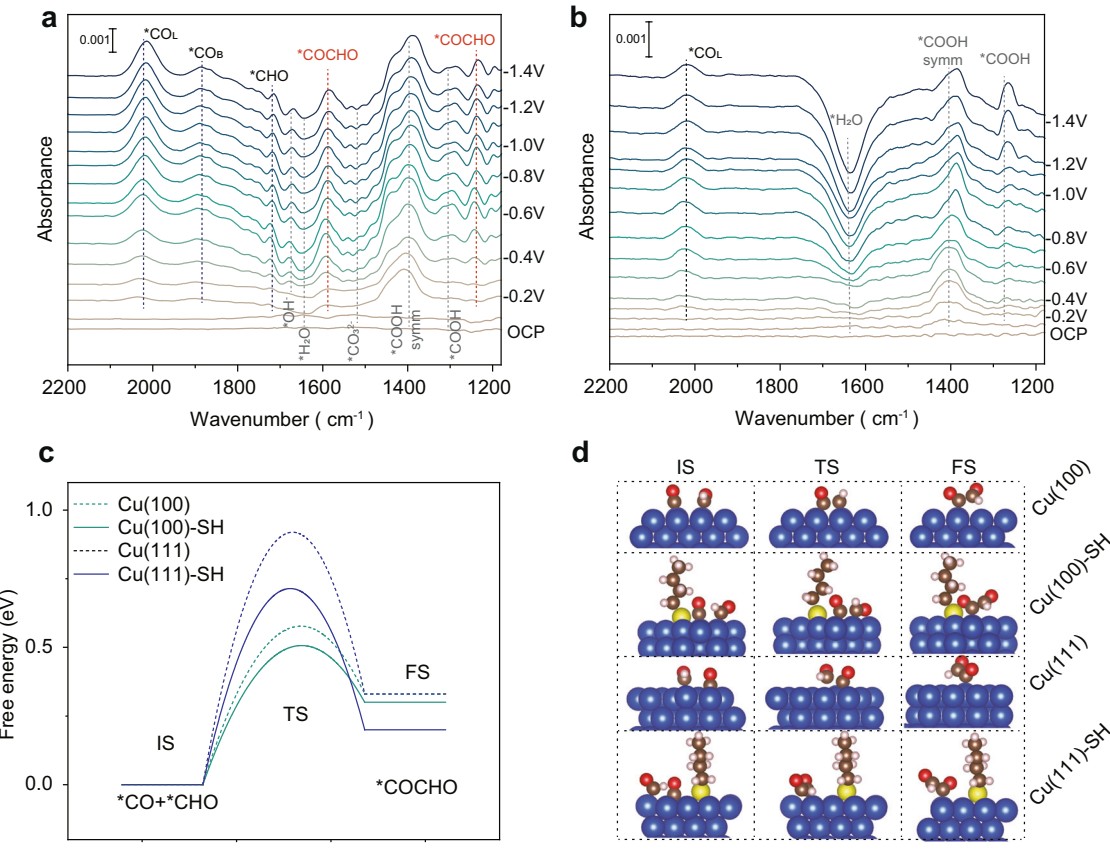

**Fig. 5 | Mechanistic studies.** In situ ATR-FTIR recorded at different applied potentials for (**a**) CuO-SH and (**b**) CuO catalysts. The free energy diagram (**c**) and optimized structures (**d**) of the CO-CHO coupling process on the Cu(100), Cu(100) with $^*C_4H_9S$, Cu(111) and Cu(111) with $^*C_4H_9S$ surfaces. Blue sphere: Cu, yellow sphere: S, brown sphere: C, pink sphere: H, red sphere: O.

Cu(111) with $^*C_4H_9S$ (Fig. 5c and Supplementary Table 5). The initial state ($^*CO$ co-adsorbed with $^*CHO$) and final state ($^*COCHO$) structures of CO-CHO coupling were determined based on our previous work[44] that has already taken into account the different adsorption sites and the steric hindrance of relative positions of $^*CO$ and $^*CHO$, as well as $^*COCHO$ (Fig. 5d). We found that the activation barrier of CO-CHO coupling on Cu(100) is 0.35 eV lower than that on Cu(111). The modification of $^*C_4H_9S$ on Cu(100) can further reduce the barrier by about 0.07 eV, leading to the improved activity and selectivity of $C_{2+}$ products, consistent with experimental observations (Fig. 2c). Moreover, the analysis of charge density difference between initial state and final state shows that thiol stabilizes the intermediate product $^*COCHO$ (Supplementary Fig. 36), consistent with the reduced free energy changes of CO-CHO coupling step on Cu surfaces with adsorbed $^*C_4H_9S$ (Fig. 5a).

In summary, we employed a hierarchical nanostructured CuO electrode functionalized with DDT to achieve superior selectivity of $CO_2RR$ for ethylene. By DDT treatment, the electrode exhibited a significant enhancement in FE of $C_2H_4$ products, reaching 72%, which is more than four times higher than the untreated electrode. Our in situ experiments results show that the DDT treatment facilitates $CO_2$ transport, enhances CO coverage on the catalyst surfaces, and stabilizes Cu(100), thus promoting the C-C coupling. Theoretical calculations further confirmed that the DDT-stabilized Cu(100) surface can effectively reduce the activation energy barrier of C-C coupling between $^*CO$ and $^*CHO$. A techno-economic analysis demonstrated promising potential for future applications (Supplementary Table 7). Our findings provide valuable insights into the mechanism of $CO_2RR$ to ethylene and offer an opportunity to develop more efficient Cu-based

electrocatalysts for the highly selective conversion of $CO_2$ to high-value-added chemicals and fuels.

## Methods
### Preparation of electrode
$Cu(OH)_2$ was grown on a piece of Cu foam (1 cm × 3 cm) through anodization following a protocol that involved washing the foam with HCl, ultrasonic cleaning in acetone, ethanol, and deionized water, and anodization at 20 mA cm$^{-2}$ for 20 min in 1 M NaOH. Annealing at 250 °C for 1 h converted $Cu(OH)_2$ nanowires to black CuO. Surface modification was achieved by submerging the CuO/Cu foam electrode in 1-dodecanethiol for 20 min at room temperature, followed by ethanol washing to remove excess 1-dodecanethiol and drying in ambient conditions by purging with compressed $N_2$.

### Materials characterization
The surface composition and valence states were analyzed with XPS, using an Escalab 250Xi X-ray photoelectron spectrometer (Thermo Fisher) with Al Ka (1486.6 eV) X-rays as the excitation source, and the binding energy of the C 1s peak at 284.8 eV was taken as an internal reference. The morphologies were examined by SEM conducted on a Hitachi SU4800 scanning electron microanalyzer with an accelerating voltage of 15 kV. Powder X-ray diffraction (PXRD) patterns were taken on PANalytical X-pert diffractometer (PANalytical, Netherlands) with Cu Kα radiation at 40 kV and 40 mA at room temperature. Transmission electron microscope (TEM) images were conducted on FEI Talos F200X G2 TEM equipment. HAADF-STEM images and energy dispersive spectra (EDS) elemental mapping were conducted on FEI Themis Z TEM equipment.

## Electrochemical measurements

Electrochemical measurements were performed using a two-compartment cell separated by a Nafion membrane (N-115, DuPont) and controlled by an electrochemical workstation (CHI660E). Each compartment contained 40 ml of 0.1 M $KHCO_3$ (Supplementary Fig. 37), with a Pt mesh (1 cm × 1 cm × 0.2 cm) and a calibrated Ag/AgCl electrode serving as the counter and reference electrodes, respectively. The working electrode area was 1 cm² (0.5 cm × 1 cm × 2 side). Prior to electrochemical measurements, $CO_2$ was continuously purged into the cathodic compartment for at least 30 min. During measurements, $CO_2$ was bubbled into the electrolyte with a constant flow rate of 20 sccm controlled by a digital mass flow controller. For the flow-cell experiment, a 1 cm² gas-diffusion electrode coated with 400 μl of well-mixed catalyst ink containing 20 μl of Nafion dispersion and cathode electrocatalyst was used as the cathode, giving a catalyst loading of 1 mg cm⁻², with Ni foam serving as the anode and Hg/HgO. The cathode catalyst is obtained by mechanical and ultrasonic exfoliation of the catalyst from the CuO or CuO-SH electrode. Two parallel fluxes of 1 M KOH were injected into the cathodic and anodic channels separated by an anion-exchange membrane. The $CO_2$ flow rate was 30 sccm, controlled by a mass flow controller. The resistance of 11.2 ± 0.1 Ω (H-cell) and 3.4 ± 0.2 Ω (flow cell) was used to calculate the iR-correction (Supplementary Fig. 39).

All potentials were calibrated to the RHE scale using the equation:

$$E(\text{RHE}) = E_{\text{applied}} + 0.059 \times \text{pH} + E_{\text{reference}}(\text{V})(E_{\text{Ag/AgCl}} = 0.197\text{V}, E_{\text{Hg/HgO}} = 0.098\text{V}) \quad (1)$$

Gas products were quantitatively analyzed using gas chromatography equipped with flame ionization and thermal conductivity detectors (Hui fen 9890E). Liquid products were collected after at least 0.5 h of electrolysis and quantitatively analyzed using 1H NMR spectroscopy with $H_2O$ suppression. The internal standard consisted of 400 μl electrolyte mixed with 50 μl dimethyl sulfoxide (20 mM) and 100 μl $D_2O$.

The FE value of a specific product was calculated based on the following equation:

$$FE_{\text{gas}} = \frac{Q_{\text{gas}}}{Q_{\text{total}}} = \frac{n_{\text{gas}}NF}{Q_{\text{total}}} = \frac{(\frac{v}{60\text{S/min}}) \times (\frac{y}{22{,}400\text{cm}^3/\text{mol}})NF}{j} \times 100\% \quad (2)$$

$$FE_{\text{liquid}} = \frac{Q_{\text{liquid}}}{Q_{\text{total}}} = \frac{n_{\text{liquid}}NF}{Q_{\text{total}}} = \frac{n_{\text{liquid}}NF}{j \times t} \times 100\% \quad (3)$$

where $v$ is the gas flow rate measured by a flowmeter, $y$ is the volume concentration of gas products, $N$ represents the number of transferred electrons for each product, $F$ denotes the Faraday constant (96,500 C mol⁻¹), $j$ signifies current, $t$ corresponds to the running time, and $n_{\text{liquid}}$ (in moles) represents the amount of liquid products determined 1H NMR.

## In situ XAFS measurements

The in situ XAS measurements were carried out using a custom-made electrochemical cell with a flat wall and a circular orifice of 15 mm diameter[45] (Supplementary Fig. 40). The catalyst-modified carbon paper was employed as the working electrode in aqueous electrolyte of 0.1 M $KHCO_3$ saturated by $CO_2$. The reference and counter electrodes were a Pt plate, and Ag/AgCl (saturated with KCl), respectively. These in situ XAS investigations were performed at the 1W2B beamline of the Beijing Synchrotron Radiation Facility (BSRF) in fluorescence mode, with the applied potential controlled by a CHI 660E electrochemical workstation. A Cu metal foil was used for calibration of the X-ray energy.

## In situ ATR-FTIR

The in situ ATR-FTIR investigations were carried out with the Bruker INVENIO spectrometer with a HgCdTe (MCT) detector cooled with liquid nitrogen. A customized electrochemical H-cell was used to collect the in situ ATR-FTIR spectra, in which Pt-wire and Ag/AgCl severed as counter and reference electrodes, respectively. A fixed-angle Si prism (60°) coated with catalysts was used as the working electrode (Supplementary Fig. 41). 0.1 M $KHCO_3$ aqueous solution constantly purged with $CO_2$ was employed as the electrolyte. All spectra were collected with 128 scans and a resolution of 4 cm⁻¹.

## In situ Raman and XRD

In situ Raman measurements were carried out by utilizing a Spectro-electrochemical flow cell. Raman measurements were conducted using a Micro-Raman spectrometer (ActonSP2500, PI) and a ×50 objective (Leica) equipped with a 785 nm laser. An Ag/AgCl electrode and a Pt plate were used as the reference and counter electrodes respectively (Supplementary Fig. 42). In situ XRD was performed at Rigaku Smart lab, and $CO_2$ electrolysis was conducted using a three-electrode electrochemical cell, with the Ag/AgCl electrode and Pt wire as the reference electrode and counter electrode, respectively (Supplementary Fig. 43). To prepare the working electrode, the catalyst ink was dropped onto a hydrophobic carbon paper. Data was collected in the electrode potential region from 0 to −1.6 V versus RHE. In situ XRD signals were collected from 30° to 60° with a scan rate of 20° min⁻¹.

## Computational methods

All the periodic DFT calculations were performed by the Vienna ab initio simulation package (VASP)[46,47]. The electron exchange correlation energy was described using a function of the revised Perdew-Burke-Ernzerhof (RPBE)[48] at Generalized Gradient Approximation (GGA) level[49]. Spin polarization was considered in the calculations, and the Methfessel−Paxton method[50] of order 2 with a smearing parameter of 0.2 eV was employed to determine the electron occupancies. Real-space projectors were used to evaluate the non-local part of the PAW potential. Additional details of the DFT calculations are provided in the Supporting Information.

## Data availability

Additional data related to this study are available from the corresponding author upon request.

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

## Acknowledgements
The project was funded by China Petroleum and Chemical Corporation under grant No. 421058. L.C. acknowledges support from the National Natural Science Foundation of China under grant No. 22203072 and the Fundamental Research Funds for the Central Universities under award 226-2022-00167. Computational resources were provided by AI + High Performance Computing Center of Institute of Computing Innovation, Zhejiang University (ZJU-ICI). Atomic-scale structural images were generated using VESTA[51].

## Author contributions
Z.C. conceived and supervised the project. Y.C.Y. performed the experiments and conducted the data analysis with contributions from J.H.W., Y.Y.F., Y.C.L., W.X.C. and Z.C. Y.C.Y. and W.X.C. performed the XAS characterizations and conducted the XAS data analysis. L.C. and T.S. designed and performed the DFT calculations. Y.C.Y. and T.S. wrote the manuscript under guidance of L.C. and Z.C. All authors discussed the results and commented on the manuscript.

## Competing interests
The authors declare no competing interests.
