## [Peer Review File · Nature Communications]

REVIEWER COMMENTS

Reviewer #1 (Remarks to the Author):

Summary

The authors develop a novel way to tailor standard Cu/CuO catalysts surfaces used in the CO₂RR to enhance the selectivity towards ethylene. They characterize their catalyst surface with a number of in-situ characterization techniques to understand the influence of DDT on their catalyst's performance. The authors show through a combination of characterization and computational modeling that their catalyst stabilizes the Cu(100) facet which is responsible for the enhanced activity. While the authors have indeed developed a catalyst that is better than the typical ones in the literature, their characterization work explaining why indeed it is more suitable would benefit from the following suggestions:-

Major Comments

Supplementary figure S9 showing higher CO₂ concentration on the CuO-SH surface vs. CuO surface is not true. The peaks for CO₂ shown here are gas phase CO₂ peaks. Those peaks can be artificially enhanced by let's say breathing around the FTIR chamber. To show an actual enhancement of CO₂ at the interface, dissolved CO₂ needs to be tracked and reported not gas phase CO₂. An explanation of how to measure gas phase CO₂ at the interface can be found in the SI of this paper.

<https://pubs.acs.org/doi/abs/10.1021/acscatal.8b01032>

The authors claim that the peak shift of CO from a HFB to LFB as the potential is made more cathodic shows that the CuO-SH has a higher coverage of more active CO. While it is true that lower frequency detected CO is more active due to the stronger binding of carbon to the copper surface, the authors should read about the concept of Stark tuning where species like CO shift to lower frequency vibrations when shifting to more negative potentials. The claim that the shift to lower frequency CO means the CO is moving to more favorable active sites is not true and is just a result of Stark tuning. If the authors want to make a claim that the CuO-SH binds CO more strongly than CuO, they should overlay traces of both at the same potential and then compare peak positions.

The authors claim that they see the presence of many of the CO₂RR intermediates in their IR spectrum. Surface enhanced IR techniques have struggled to detect these intermediates so seeing them so clearly with regular ATR-FTIR is surprising. With regards to that, the CuO sample shows big OH bending modes in the 1600-1700 cm⁻¹ region but their presence is not seen in the CuO-SH sample. How were those water peaks subtracted out to reveal intermediates below them? Solution phase carbonate and bicarbonate show up in the 1300-1440 cm⁻¹ region. How are the authors sure that their peak assignments in those regions are not solution phase carbonate and bicarbonate peaks? Do the peaks

assigned to the intermediates shift with potential (show Stark tuning) which is a way to differentiate surface bound vs. solution phase peaks?

Minor Comments

Supplementary figure 4 shows ATR-FTIR traces that were used to prove successful functionalization of the CuO surface. The authors do not specify how the surface was treated after the functionalization was complete and before the ATR-FTIR spectra were recorded. Was the CuO pulled out of the DDT solution and then placed directly in the FTIR chamber? If that is the case, I would expect traces of the DDT solution on the CuO surface to give a signal for CH₃ and CH₂ stretching modes. Later in the methods section, they claim that the surface is washed with ethanol and then air dried. Are there traces of leftover ethanol on the surface that lead to these CH₃ and CH₂ peaks being seen? What would be more convincing to see is the traces of pure CuO and then CuO after being functionalized and cleaned to remove traces of DDT and ethanol solution to show a permanent functionalization. I have a similar concern for the other characterization methods like EDS where the lack of explanation of the cleaning methods reduces the credibility of the results. Later, in the post electrochemical reaction analysis of the catalyst's surface with EDS still seeing the functionalization is a bit more reassuring.

Does 0.1 M KHCO₃ in the H-cell provide enough ion transport? I would expect electrolytes at such low concentration to provide a high internal solution resistance (potential drop between the working and reference electrode)? I am not sure we can accurately trust the potential vs. RHE number if we don't have a sense for the internal resistance in the H-cell used. Have higher concentrations of KHCO₃ been used, why or why not?

Can the authors explain why the electrolyte was changed from K₂CO₃ to KOH for the GDE experiment? Was the GDE experiment probing CORR? It isn't clear! KOH electrolyte will not work well for long term CO₂RR.

The Raman spectra do show a more intense adsorbed CO peak in the CuO-SH case vs. the CuO case. While to the naked eye this looks the case, the authors do claim to normalize the intensity to the glassy carbon peak which comes from the covering film of the electrochemical cell. Can the authors show the peak intensity of this glassy carbon peak is similar between the CuO and CuO-SH run to give the reader confidence that these increases in peak intensity for other peaks such as CO are not a setup dependent phenomenon? Can the higher peak intensity of the CuO-SH sample be because there are more active sites exposed in the area the Raman laser hits and not just because of a higher coverage? Have the authors moved the laser beam to different sites on the catalyst surface to see if peaks are as intense there too?

The authors claim that their IR measurements show that CuO-SH has bridge bonded CO while CuO does not. While it is true that bridge bonded CO is a more strongly bounded version of CO, the consensus in the field is that bridge bonded CO is not active in the CO₂RR. Hence, its presence does not enhance the CO₂RR activity.

It is extremely commendable that the authors have been able to extensively characterize their catalyst surface with so many in-situ techniques. I would recommend attaching photographs or schematics of the setups along with the description especially for the in-situ XRD and in-situ XAS as these setups are not common and usually custom built for electrochemical reaction studies. The field would benefit with this knowledge tremendously. The authors have already shown their IR and Raman setups in the SI which is good to see.

Reviewer #2 (Remarks to the Author):

Yao et al. designed a copper-(Cu-) based catalyst to enable a highly selective electrocatalytic conversion from CO₂ to ethylene. The authors claims that a thiol ligand on the CuO precursor can preferentially stabilize the Cu(100) facet as the CuO reduces into metallic Cu during the CO₂ reduction reaction, and that it is the stabilized Cu(100) facet that enables the high ethylene selectivity. The authors report FEs as high as ~ 72% (in an H-cell) and ~79% (in a flow cell).

The reviewer appreciate the high ethylene FE achieved in the present work, it is for sure among the top of existing works although not record-breaking. However, the reviewer questions the mechanism as the authors present, the stability of the Cu(100) surface and the techno-economic feasibility of the technique. These concerns prevent the reviewer from recommending publication of the work in Nature Communications.

1. I appreciate that the authors have conducted operando XRD to investigate the preferred Cu facet during CO₂RR. In fact this is the only direct evidence of the facet preference in the work. Since the Cu(100) facet, and its stabilization, is of critical importance of the work, I feel that an in-situ XRD alone is not sufficient to form a solid chain of characterization evidence. The authors concluded that the CuO-SH-derived catalyst exhibit a higher intensity for the Cu(100) peak. This may be true, but it is also clear that the Cu(111) peak still dominates even for the CuO-SH-derived catalyst, which may infer the domination of Cu(111) on the catalyst surface. In this case, can the authors attribute the high ethylene FE to the Cu(100) facet if it is still a minority facet on the CuO-SH-derived catalyst surface?

2. In Fig. 2c and d, the authors attempt to attribute the higher ethylene FE to a stronger *CO adsorption (characterized by a red-shift of an atop *CO peak). I have a related question and a minor comment. First, how does the authors attribute the stronger *CO adsorption to the thiol ligand? It may as well be just from the increased over-potential. The same data as in Fig. 2d should be provided for the CuO without thiol. Secondly, I suggest the authors to use a brighter color for the peaks in Fig. 2d, the peaks look like dips as it is currently colored.

3. Along this thread, if the authors would like to design a catalyst with a majority of Cu(100) surface to start with (a Cu cube for example, see *Angew. Chem. Int. Ed.* 2016, 55, 5789–5792), and stabilize the surface using the thiol?

4. Even one is to accept Cu(100)'s role in the high FE, whether it is “stabilized” on the catalyst surface by the thiol ligand is a question. The stability test is run only for 5 hours, and during that 5 hours the FE is already decreased from 72% to 60%. To compare, stability in electrochemical ethylene synthesis has been reported in the hundred-hour range for years (for example, *Science* 360, 783–787 (2018), *Nature*, 577(7791), 509-513). What leads to a decrease of ethylene FE? Has surface reconstruction taken place? Has the Cu(100) surface been destabilized? Can the authors do some characterization, an operando XRD for example, to find out? Also, since the authors claim that the key to the high ethylene selectivity is the “stabilized Cu(100)” as they put into the title, the stability test will be of core importance and it deserves a main-text figure.

5. Talking about the performance, I appreciate a high FE, but I am also concerned about the high over-potential where the high FE is achieved. In other words, a high current efficiency may be achieved but at the cost of a low voltage efficiency (the energy efficiency can be thought of as the product of both efficiencies). This can be understood from the operando XRD: the Cu(100) facet is only promoted at a high potential (although I don't understand why). I encourage the authors to consider the energy efficiency, and perform a techno-economic analysis to evaluate the application potential, of their technique.

6. Along this thread: the authors achieve an all-round better performance in a flow cell than in an H-cell (higher FE, much higher current density, lower over-potential). Why didn't the authors present the flow-cell-performance in the maintext and leave the H-cell-performance in the SI?

A few other comments:

7. In Fig. 3, why do the authors choose different x-domain for the Raman shifts (Panels a, b and d)?

8. Fig. 2, we can see that a major reason for a low ethylene FE for the CuO is the higher HER. What leads to it? Is it again related to the Cu(111) facet? There should be a good explanation.

9. Along this thread: Fig. 2 should emphasize more on the C₂₊:C₁ ratio because boosting C-C coupling, instead of HER suppression, is the major theme.

Reviewer #3 (Remarks to the Author):

In this work, the authors developed a hierarchical nanostructured CuO catalyst stabilized with DDT for CO₂ electrochemical reduction to ethylene. The catalyst was evaluated in both H-cell and flow cell for electrocatalytic performance. The Faradaic Efficiency of C₂H₄ reached up to 72%. Based on the catalytic performance, catalyst characterizations, and theoretical calculations, the authors claimed that DDT treatment of CuO catalyst could stabilize Cu(100) facets and promote the C-C coupling, which leads to high C₂H₄ selectivity and good catalyst stability.

This work aimed to address the great challenges in CO₂RR toward C₂ products and the obtained C₂H₄ selectivity (FE) is among the best reported results in literature. Using thiol to stabilize Cu(100) facets provides a promising route to develop new CO₂RR catalysts toward C₂ products with high selectivities and will benefit the advancement of CO₂RR toward industrial applications. The manuscript was well organized and written. Most of the catalyst characterization, DFT calculation, and catalyst performance evaluations were well designed and support the claims. However, the current work lacks key experiments and characterizations for the CuO-SH stability performance. Therefore, this manuscript is not recommended for acceptance at current version and review is needed after revision.

Below are the detailed comments:

1) What was the DDT coverage on CuO surface? What was the Thiol/CuO ratio? Any thiol coverage effects on catalytic performance?

2) Based on 5 hours stability tests, it seems that CuO-SH showed a significant performance degradation with a descending trend, although the rate, as claimed by authors, was slower (16%) than that of CuO (48%) in 5 hours. Any interpretation of what causes such performance degradation of CuO-SH?

3) Did authors observe any structure changes of CuO-SH after stability testing? XRD results after stability testing?

4) It seemed the authors did stability testing in an H-cell at a constant voltage for 5 hours. Usually, a 5-hour test is too short for an electrode catalyst stability evaluation. Since the authors also checked CuO-SH performance in a flow cell, it will be highly desirable to run the stability testing in the flow cell under constant current operation mode (200 or 300 mA cm⁻² current density) for a longer time, at least several tens of hours. Post characterizations of thiol, and possible structure changes should be conducted after such long-term testing. This will be critical to evaluate the potential application of the proposed thiol-catalyst system regarding the industrial application requirements.

5) For the ECSA measurements, did the authors do the IR correction? The potential scan range (0.3 V) was much wider than the normal one (0.1V) used for such measurements. I suggest running the potential scan between 0.175 V and 0.275V.

6) Line 379, "Nafion membrane (N-115, dubant)", please clarify "dubant".

7) Line 382-383, "working electrode area was 1 cm² (0.5 cm x 1 cm x2)," please clarify part in the brackets.

8) Line 388-395, please clarify those in the brackets "(400 uL) of well-mixed catalyst ink containing (20 mL) of Nafion dispersion", please also clarify how the 1 M KOH electrolyte circulated during the flow cell test.

9) Line 391, "1 M KOH" was used for flow cell testing, in the caption of Supplementary Fig. 8, "0.1 M KHCO₃ electrolyte", please clarify.

10) Line 415-417, this may cause confusion since the sentence is for working electrode treatment while the supplementary Fig.17 is the photo of the in situ ATR-FTIR setup.

Point-by-point Responses to Reviewers' Comments

Reviewer #1 (Remarks to the Author):

Summary Comment: The authors develop a novel way to tailor standard Cu/CuO catalysts surfaces used in the CO₂RR to enhance the selectivity towards ethylene. They characterize their catalyst surface with a number of in-situ characterization techniques to understand the influence of DDT on their catalyst's performance. The authors show through a combination of characterization and computational modeling that their catalyst stabilizes the Cu(100) facet which is responsible for the enhanced activity. While the authors have indeed developed a catalyst that is better than the typical ones in the literature, their characterization work explaining why indeed it is more suitable would benefit from the following suggestions:

Our response: We thank the reviewer for the positive comments, and appreciate the thoughtful and constructive suggestions to improve the overall quality of the work. All these insightful suggestions raised by the reviewer have been thoroughly considered and the corresponding revisions have been made as below.

Major Comment 1: Supplementary figure S9 showing higher CO₂ concentration on the CuO-SH surface vs. CuO surface is not true. The peaks for CO₂ shown here are gas phase CO₂ peaks. Those peaks can be artificially enhanced by let's say breathing around the FT-IR chamber. To show an actual enhancement of CO₂ at the interface, dissolved CO₂ needs to be tracked and reported not gas phase CO₂. An explanation of how to measure gas phase CO₂ at the interface can be found in the SI of this paper. <https://pubs.acs.org/doi/abs/10.1021/acscatal.8b01032>

Our response:

We thank the reviewer for the valuable comments. Indeed, our previous tests overlooked the impact of gaseous CO₂ in the FT-IR chamber. Following the reviewer's suggestion, we have retested the localized CO₂ concentrations of the CuO and CuO-SH electrodes, using the experimental methods reported in

the literature [*ACS.Catal.* **8**, 3999–4008 (2018)]. The specific testing procedure is as follows: we prepared two electrolyte solutions: Ar-saturated 0.1M KHCO₃ and CO₂-saturated 0.1M KHCO₃. At first, we collected the background spectra using the Ar-saturated 0.1M KHCO₃ electrolyte solution. Next, we collected the infrared spectra of the CuO and CuO-SH electrodes at the open-circuit potential in the CO₂-saturated 0.1M KHCO₃ electrolyte solution. From Fig. R1, it can be observed that the CuO-SH electrode exhibits a higher localized CO₂ concentration at the interface. This enhancement can be attributed to its superhydrophobic surface, which promotes the capture of carbon dioxide at the interface.

Fig R1. CO_{2(aq)} spectra of CuO-SH and CuO collected at open circle potential in CO₂ saturated 0.1M KHCO₃.

Our revision: According to the reviewer's comment, the related figure has been added in the revised version. Please see the yellow highlighted parts on page 14 in the revised Supplementary Information.

Major Comment 2: The authors claim that the peak shift of CO from an HFB to LFB as the potential is made more cathodic shows that the CuO-SH has a higher coverage of more active CO. While it is true that lower frequency

detected CO is more active due to the stronger binding of carbon to the copper surface, the authors should read about the concept of Stark tuning where species like CO shift to lower frequency vibrations when shifting to more negative potentials. The claim that the shift to lower frequency CO means the CO is moving to more favorable active sites is not true and is just a result of Stark tuning. If the authors want to make a claim that the CuO-SH binds CO more strongly than CuO, they should overlay traces of both at the same potential and then compare peak positions.

Our response:

We thank the reviewer for your meaningful suggestion. When an external electric field is applied, the center of characteristic peaks tends to undergo a shift and the magnitude of this Stark-induced tuning typically varies with the strength of the electric field. As suggested by the reviewer, in order to minimize the influence of the Stark effect, we have overlapped the CuO and CuO-SH spectra at the same potential, as shown in Fig R2. By comparing the peak positions of the CO adsorption peaks, we observe that within this potential range, the CO adsorption peak of CuO-SH is slightly lower than that of CuO, indicating a stronger CO binding ability of CuO-SH. In addition, the *in situ* infrared spectroscopy experiments show that the spectrum of CuO-SH exhibits a stronger *CO peak than that of CuO (see Fig. 5a and 5b in our revised manuscript). Both *in situ* Raman and *in situ* infrared results provide solid evidence to confirm that the DDT modification enhances the CO binding capability of the CuO-SH catalyst.

Fig. R2 *In situ* Raman spectra of CuO-SH and CuO were collected within a potential window of -1 to -1.4 V in CO₂ saturated 0.1M KHCO₃.

Our revision: According to the reviewer's comments, the corresponding revisions have been made. Please see the yellow highlighted part on pages 10-11 (lines 235-238, 242-243) in the revised manuscript.

Major Comment 3 : The authors claim that they see the presence of many of the CO₂RR intermediates in their IR spectrum. Surface enhanced IR techniques have struggled to detect these intermediates so seeing them so clearly with regular ATR-FTIR is surprising. With regards to that, the CuO sample shows big OH bending modes in the 1600-1700 cm⁻¹ region but their presence is not seen in the CuO-SH sample. How were those water peaks subtracted out to reveal intermediates below them? Solution phase carbonate and bicarbonate show up in the 1300-1440 cm⁻¹ region. How are the authors sure that their peak assignments in those regions are not solution phase carbonate and bicarbonate peaks? Do the peaks assigned to the intermediates shift with potential (show Stark tuning) which is a way to differentiate surface bound vs. solution phase peaks?

Our response:

We appreciate your professional comments. We believe that our clear IR

spectrum benefits from the following reasons: 1) The relatively strong signal of the intermediates for CuO-SH compared to CuO; 2) The accumulation of a large number of repetitive scans enhances the signal; 3) Our IR equipment with a liquid nitrogen cooled HgCdTe detector is beneficial for collecting weaker signals.

The peak near 1600-1700 cm^{-1} mentioned by the reviewer, corresponding to the O-H vibration of water, was observed in both CuO-SH and CuO catalysts. We did not subtract or suppress the water peak in our analysis. Due to the specific configuration of *in situ* infrared electrolytic cell, there is a certain degree of mass transfer restriction, which results in a downward peak near 1600-1700 cm^{-1} , corresponding to the O-H vibration of water. Similar results have been reported in other studies as well. [*Nat Commun.* **13**, 3754 (2022); *Angew. Chem. Int. Ed.* **134**, e2022049 (2022); *Angew. Chem. Int. Ed.* **133**, 25689–25696 (2021); *Angew. Chem. Int. Ed.* **61**, e202209268 (2022)]

The reviewer raised concerns about the attribution of the characteristic peaks of the intermediates detected in the 1300-1400 cm^{-1} region due to the close proximity of the peaks of the carbonate and bicarbonate in the solution phase. Here, we clarify the question in detail as follows:

1) In the *in situ* infrared experiment, background spectra were collected in a CO_2 -saturated 0.1M KHCO_3 electrolyte at 0.2V. The subtraction of the background spectra aimed to minimize the influence of species like HCO_3^- and CO_3^{2-} in the solution on the characteristic peak signals.

2) As suggested by the reviewer, we employed Stark tuning to identify the peaks assigned to the intermediates. In Fig. R3a and 3b, we observed a significant stark induced-tuning at a characteristic peak around 1400 cm^{-1} with increasing overpotential, which indicates that the peaks can be attributed to surface-bound intermediates. Similar results can be found in numerous reports, which attribute the characteristic peak at 1400 cm^{-1} to $^*\text{COOH}$. [*J. Am. Chem. Soc.* **139**, 15664–15667 (2017); *Angew. Chem.* **133**, 25689–25696 (2021); *Angew. Chem.* **61**, e20211408 (2022); *Nat. Commun.* **13**, 63 (2022); *J. Am. Chem. Soc.* **143**,

7242–7246 (2021)].

3) In addition, to further confirm the attribution of the characteristic peak around 1400 cm^{-1} , we selected N_2 -saturated 0.1 M KHCO_3 electrolyte and collected the *in situ* IR spectra of CuO-SH at the potential range of $0 \sim -0.8\text{ V}$ vs RHE with 0.2 V spectra as the background. As shown in Fig. R3c, the peak located near 1400 cm^{-1} was not detected at the same potential in the CO_2 -free electrolyte; therefore, we believe that the detected peak in CO_2 -saturated 0.1 M KHCO_3 electrolyte is attributed to the $^*\text{COOH}$ intermediate adsorbed on the catalyst surface in the CO_2RR reaction.

Fig. R3. *In situ* ATR-FTIR recorded at different applied potentials for **a** CuO and **b** CuO-SH catalysts within a potential window of 0 to -0.8 V in CO_2 saturated 0.1 M KHCO_3 , **c**, *in situ*

ATR-FTIR of CuO-SH collected in N₂-saturated 0.1M KHCO₃.

Our revision: According to the reviewer's comments, the corresponding revisions have been made. Please see the yellow highlighted part on page 26 in the revised supplementary information.

Minor Comment 1: Supplementary figure 4 shows ATR-FTIR traces that were used to prove successful functionalization of the CuO surface. The authors do not specify how the surface was treated after the functionalization was complete and before the ATR-FTIR spectra were recorded. Was the CuO pulled out of the DDT solution and then placed directly in the FTIR chamber? If that is the case, I would expect traces of the DDT solution on the CuO surface to give a signal for CH₃ and CH₂ stretching modes. Later in the methods section, they claim that the surface is washed with ethanol and then air dried. Are there traces of leftover ethanol on the surface that lead to these CH₃ and CH₂ peaks being seen? What would be more convincing to see is the traces of pure CuO and then CuO after being functionalized and cleaned to remove traces of DDT and ethanol solution to show a permanent functionalization. I have a similar concern for the other characterization methods like EDS where the lack of explanation of the cleaning methods reduces the credibility of the results. Later, in the post electrochemical reaction analysis of the catalysts surface with EDS still seeing the functionalization is a bit more reassuring.

Our response:

Thank you for carefully reviewing. Indeed, the relevant details are missing in our manuscript. After surface modification, the CuO-SH electrode undergoes several ethanol washes and then is dried by purging with compressed N₂ before conducting the ATR-IR testing and other characterizations. For example, the CuO-SH samples for EDS analysis and STEM were obtained by sonication of the ethanol-washed CuO-SH electrode. Considering the concerns raised by the reviewers regarding the possibility of residual ethanol on the electrode surface due to the washing procedure, we performed the ethanol washes treatment on

CuO. The ATR-IR results of the CuO sample washed with ethanol did not show the presence of C-H bonds(Fig. R4).

Fig. R4. FI-IR spectra of the CuO-SH washed by ethanol after surface modification, ethanol-washed CuO and liquid DDT.

Our revision: The figure and related experimental details have been added to the revised supplementary information. Please see the yellow highlighted part on page 6 in the revised supplementary information.

Minor Comment 2: Does 0.1 M KHCO₃ in the H-cell provide enough ion transport? I would expect electrolytes at such low concentration to provide a high internal solution resistance (potential drop between the working and reference electrode)? I am not sure we can accurately trust the potential vs. RHE number if we don't have a sense for the internal resistance in the H-cell used. Have higher concentrations of KHCO₃ been used, why or why not?

Our response:

We appreciate the reviewer for the valuable comments. To evaluate the effect of electrolyte concentration on internal resistance, we tested their EIS spectra at 0.1 M, 0.25 M and 0.5 M, respectively. From the EIS data shown in Fig. R5a, the solution resistance values for different KHCO₃ concentrations are as follows: 11.9 ohms for 0.1 M, 10.61 ohms for 0.25 M, and 8.8 ohms for 0.5 M. As the

concentration increases, the resistance slightly decreases. To make the potential vs. RHE number reliable, we will conduct a solution resistance test prior to each performance test and apply iR compensation to ensure accurate potential measurements.

Furthermore, we evaluated the catalytic performance of the CuO-SH catalyst in the H-cell using three different concentrations of KHCO_3 : 0.1 M, 0.25 M and 0.5 M, respectively. As shown in Fig. R5b, increasing the KHCO_3 concentration resulted in a decrease in the Faraday efficiency of C_2H_4 production and an increase in the Faraday efficiency of the HER.

Usually, HCO_3^- in the electrolyte plays several roles, including ion conduction, providing protons, and transporting CO_2 by balance reaction. Therefore, increasing the bulk concentration of HCO_3^- not only enhances ion conductivity but also leads to an increase in the competing HER rate and a decrease in localized CO_2 concentration [*Acc. Chem. Res.* **55**, 1900–1911 (2022); *J. Am. Chem. Soc.* **139**, 15664–15667 (2017); *J. Appl. Electrochem.* **36**, 161–172 (2006)]. Therefore, based on the above experimental results and analysis, we believe it is reasonable to select 0.1 M KHCO_3 solution as the electrolyte as reported in most of the literatures.

Fig. R5. **a**, EIS of CuO-SH electrode in 0.1 M KHCO_3 and 0.5 M KHCO_3 , **b**, Faradaic efficiencies (FE) of the H_2 and C_2H_4 over CuO-SH in 0.1 M, 0.25 M and 0.5 M KHCO_3 , respectively.

Our revision: The related modifications have been made. Please see the yellow highlighted part on page 19 (lines 445-446) in the revised manuscript

and page 34 in the revised supplementary information.

Minor Comment 3: Can the authors explain why the electrolyte was changed from K₂CO₃ to KOH for the GDE experiment? Was the GDE experiment probing CORR? It isn't clear! KOH electrolyte will not work well for long term CO₂RR.

Our response:

Thank you for the helpful comments. Usually, in an H-cell the long diffusion distance poses a challenge as CO₂ and OH⁻ tend to form HCO₃⁻/CO₃²⁻, hindering the transportation of CO₂ to the reaction site. Consequently, the KOH electrolyte is not suitable for H-cells. Conversely, neutral KHCO₃ plays a role in providing protons and facilitating the transfer of carbon dioxide. Hence, neutral KHCO₃ often is chosen as the electrode solution for the H-Cell test. In a flow cell setup, CO₂ gas can diffuse to the catalyst surface through the gas diffusion channel of the gas diffusion electrode (GDE). This configuration significantly reduces the diffusion distance and ensures a sufficient supply of CO₂. Moreover, alkaline KOH electrolytes have a significant impact on the selectivity of C₂⁺ products in the CO₂RR reaction, enhancing their formation, and reducing the onset potential for C₂H₄ generation. Additionally, the anodic oxygen evolution reaction (OER) exhibits faster kinetics in alkaline conditions. Therefore, choosing alkaline electrolytes not only enhances the selectivity and efficiency of C₂⁺ products but also improves the overall energy conversion efficiency. Therefore, 1M KOH can be chosen as the electrolyte for the flow cell setup.

The purpose of our GDE experiment was not to explore CORR, and it is not that KOH electrolytes do not work well for long-term CO₂RR. Our purpose of GDE experiments is to achieve highly selective ethylene conversion in the flow cell so that we can demonstrate its application prospects.

Minor Comment 4 : The Raman spectra do show a more intense adsorbed CO peak in the CuO-SH case vs. the CuO case. While to the naked eye this looks

the case, the authors do claim to normalize the intensity to the glassy carbon peak which comes from the covering film of the electrochemical cell. Can the authors show the peak intensity of this glassy carbon peak is similar between the CuO and CuO-SH run to give the reader confidence that these increases in peak intensity for other peaks such as CO are not a setup dependent phenomenon? Can the higher peak intensity of the CuO-SH sample be because there are more active sites exposed in the area the Raman laser hits and not just because of a higher coverage? Have the authors moved the laser beam to different sites on the catalyst surface to see if peaks are as intense there too?

Our response:

Thank you for your comments. In response to the reviewer's concern about the effect of the test setup on the CO peaks, the *in situ* Raman experiments were performed in the same setup, as shown in Fig. R6, the peak intensities of the glassy carbon electrode in the *in situ* Raman spectra of CuO-SH and CuO were found to be similar.

Furthermore, the reviewer was worried about the effects of the exposure active sites on CO peaks. In fact, the unmodified CuO catalyst exhibits a larger ECSA under the same catalyst loading conditions (as shown in Fig. R23). Thus, we can rule out the possibility that the stronger CO peaks in the Raman spectra of CuO-SH originate from exposing more active sites.

As regarding the consistency of the signal intensity at the catalyst surface, we selected three different positions on the surface of the glassy carbon electrode loaded with CuO-SH catalysts to collect the Raman spectra. As depicted in Fig. R7, the spectral intensities obtained from the three positions demonstrated remarkable similarity, demonstrating good consistency of the sample.

Fig. R6 *In situ* Raman spectrum of CuO and CuO-SH in the characteristic peak range of glass carbon at the potential range of 0 ~-1.4V vs RHE.

Fig. R7 *In situ* Raman spectrum of CuO-SH collected at three distinct positions at the potential range of -0.2 ~-1.4V vs RHE.

Our revision: The related modification has been made. Please see the yellow highlighted part on page 10 (line 231) in the revised manuscript and page 15 in the revised supplementary information.

Minor Comment 5 : The authors claim that their IR measurements show that CuO-SH has bridge bonded CO while CuO does not. While it is true that bridge bonded CO is a more strongly bounded version of CO, the consensus in the

field is that bridge bonded CO is not active in the CO₂RR. Hence, its presence does not enhance the CO₂RR activity.

Our response:

We thank the reviewer for the valuable comments, which improve our understanding of mechanisms. Although there is still controversy about whether bridge bonded CO has activity (non-active reports: [*ACS Catal.* **8**, 7507–7516 (2018); *Nat. Commun.*, **13**, 2656 (2022); *Nat. Catal.*, **3**, 775–786 (2022)]; active reports: [*J. Am. Chem. Soc.* **142**, 2857–2867 (2020); *Nat. Commun.*, 2022, 13, 3754 (2022); *Science Bulletin* **66**, 62–68 (2021)], it must be frankly stated that our previous analysis based solely on literature reports was not rigorous enough.

To be honest, it is still a great challenge to distinguish which adsorption configuration is active by experiment. Based on our *in situ* IR results, it is apparent that the CuO-SH catalysts with an increased proportion of Cu(100) facets exhibit a predominant occurrence of top-adsorbed CO as the primary adsorption configuration. Moreover, the intensity of top-adsorbed CO was notably enhanced in comparison to that in the unmodified CuO catalysts (see Fig.5a and 5b in the revised manuscript). To figure out which configuration is active, we performed computational calculations to identify the most favorable CO adsorption structure on Cu(100) surfaces spanning a range of CO coverages, specifically from 1/16 to 8/16. As depicted in Figure R8, the top-adsorbed CO configuration emerged as the optimal one on Cu(100), exhibiting a binding energy ranging from -1.18 to -1.10 eV across the coverage range of 1/16 to 8/16. Furthermore, the presence of *CHO species adsorbed on CuO-SH is observed by our *in situ* IR spectra, which arises as a consequence of *CO protonation and serves as a crucial intermediate in C–C asymmetric coupling. (See Fig. 5a in the revised manuscript, peak located at ~1720 cm⁻¹). According to our DFT results in Figure R9, we found that the top-adsorbed CO configuration exhibits more favorable protonation reaction barriers in comparison to the bridge-adsorbed CO configuration on Cu(100). The

increased proportion of Cu(100) facets promoted the coverage of active top-adsorbed CO on CuO-SH surface.

Thus, combining the results of our DFT calculations with the *in situ* IR spectra, we can infer, as the reviewer suggested, that the activity of bridge-adsorbed CO is weaker than that of top-adsorbed CO. We have adjusted the description of bridge-adsorbed CO in the revised manuscript.

Fig. R8, The lowest CO absorption energy on Cu(100) across a range of CO coverages from 1/16 to 8/16

Fig. R9, The optimized structures (left) and reaction barriers (right) of top/bridge-adsorbed CO protonation process on the Cu(100) facets.

Our revision: The related modification has been made in the revised manuscript. Please see the yellow highlighted part on page 15 (lines 352-355, 361-363) in the revised manuscript as well as page 27 in the revised supplementary information.

Minor Comment 6 : It is extremely commendable that the authors have been able to extensively characterize their catalyst surface with so many in-situ techniques. I would recommend attaching photographs or schematics of the setups along with the description especially for the in-situ XRD and in-situ XAS as these setups are not common and usually custom built for electrochemical reaction studies. The field would benefit with this knowledge tremendously. The authors have already shown their IR and Raman setups in the SI which is good to see.

Our response:

Thank you for your recognition of our work. The following figures have been added to the revised Supplementary Information.

Fig. R10 Photograph of *in situ* XRD setup.

Fig. R11 Photograph of *in situ* XAFS setup.

Our revision: The figures have been added in the revised supplementary information. Please see the yellow highlighted part on pages 37-38 the revised supplementary information.

Reviewer #2 (Remarks to the Author):

General Comment: Yao et al. designed a copper-(Cu-) based catalyst to enable a highly selective electrocatalytic conversion from CO₂ to ethylene. The authors claims that a thiol ligand on the CuO precursor can preferentially stabilize the Cu(100) facet as the CuO reduces into metallic Cu during the CO₂ reduction reaction, and that it is the stabilized Cu(100) facet that enables the high ethylene selectivity. The authors report FEs as high as ~ 72% (in an H-cell) and ~79% (in a flow cell).

The reviewer appreciate the high ethylene FE achieved in the present work; it is for sure among the top of existing works although not record-breaking. However, the reviewer questions the mechanism as the authors present, the stability of the Cu(100) surface and the techno-economic feasibility of the technique. These concerns prevent the reviewer from recommending publication of the work in Nature Communications.

Our response:

We appreciate the reviewer for the positive comments and thank the thoughtful and important suggestions to improve the overall quality of the work. We will address these questions as below.

Comment 1: I appreciate that the authors have conducted operando XRD to investigate the preferred Cu facet during CO₂RR. In fact, this is the only direct evidence of the facet preference in the work. Since the Cu(100) facet, and its stabilization, is of critical importance of the work, I feel that an in-situ XRD alone is not sufficient to form a solid chain of characterization evidence. The authors concluded that the CuO-SH-derived catalyst exhibit a higher intensity for the Cu(100) peak. This may be true, but it is also clear that the Cu(111) peak still dominates even for the CuO-SH-derived catalyst, which may infer the domination of Cu(111) on the catalyst surface. In this case, can the authors attribute the high ethylene FE to the Cu(100) facet if it is still a minority facet on the CuO-SH-derived catalyst surface?

Our response:

We appreciate the reviewer for this valuable suggestion.

(1) To be honest, currently, only *in situ* X-ray diffraction is a relatively convenient technique. The complexity of *in situ* electrochemistry and catalyst structures limits further characterization of their microstructure through techniques such as *in situ* transmission electron microscopy (TEM) or scanning transmission electron microscopy (STEM). In-situ XRD is a powerful technique for obtaining statistically averaged bulk crystal structures of catalysts under *in situ* conditions, which is beneficial for understanding the overall changes of crystal structure in the sample.

According to the reviewer's comments, we conducted an OH⁻ electro-adsorption experiment to further analyze the surface structure of CuO and CuO-SH after 30 min CO₂RR at the potential ranged from -1 to -1.4V. Different copper facets exhibit distinct OH⁻ electrochemical adsorption behavior. [*J. Electroanal. Chem.* **112**, 387–390 (1980)]. Typically, Electrochemical OH⁻ adsorption was performed in an N₂-saturated 1 M KOH electrolyte with a linear sweep voltammetry method at a sweep rate of 100 mV s⁻¹. The potential ranged from -0.2 to 0.6V versus RHE.

As shown in Fig. R12, linear sweep voltammetry profiles reveal electrochemical OH⁻ adsorption peaks on Cu(100), Cu(110) and Cu(111) at potentials ~0.36, 0.42, 0.47 V versus RHE, respectively. By integrating the OH⁻ adsorption peak of Cu(100) and Cu(111), we can obtain the relative ratio of the Cu(100)/Cu(111) in the crystal surface of CuO-SH and CuO. In the OH⁻ adsorption analysis, we observed a similar Cu(100)/Cu(111) ratio for both the CuO and CuO-SH catalysts, which is in agreement with the results of the *in situ* XRD measurements(Fig. R13). At potentials ranging from -1 to -1.4V vs RHE, we observed a high Cu(100)/Cu(111) ratio of 1.16, 1.04, and 1.09, respectively. Both OH adsorption experiments and *in situ* XRD confirm the promotion of Cu(100) by DDT in the CuO-SH catalyst.

(2) As regarding whether the Cu(100) facet is the active crystalline surface for

ethylene generation, we infer that the main reason why the Cu(100) facet is active is based on our following experimental observations: Although the Cu(100) facet does not exceed the Cu(111) surface (as potential increases, the facet ratio of Cu(100)/(111) approach 1), the ethylene selectivity increases with the increase of Cu(100) surface. Let's assume that the Cu(111) facet is a more active crystal plane for ethylene formation. The CuO sample with a higher (111) proportion than CuO-SH (as shown in Fig. 4f) should have higher ethylene selectivity. However, the experimental results are the opposite of this.

To reveal which crystal plane has higher activity for ethylene selectivity, we selected well-defined Cu(100) and Cu(111) single crystal foil as a simplified model for eliminating the influence of other factors. We use polycrystalline Cu foils to obtain single crystal Cu(100) and Cu(111) foils through different annealing treatments according to literature methods [*Chem.* **7**, 406-420 (2021)]. The XRD results in Figure R14a show their single crystal characteristics of Cu(100) and Cu(111) foils. The corresponding CO₂RR performances indicate that although the ethylene conversion efficiency of both single crystal foils is not high, Cu(100) definitely exhibits a higher ethylene FE than Cu(111) (Fig. R14).

To understand the impact of catalyst facets on ethylene selectivity under real condition, we further investigated the time-dependent ethylene selectivity with structural evolution of CuO-SH. We collected the time-resolved *in situ* XRD of CuO-SH at -1.4 V in 0.1M KHCO₃ and also carried out the time-dependent performance testing over CuO-SH electrode at -1.4 V. To observe the transformation process of crystal planes, it is worth noting that the CuO-SH samples used in this test were not subjected to an activation treatment. As shown in Fig. R15, the diffraction peak of Cu(100) gradually increases within the first 40 minutes, and the ratio of Cu(100): Cu(111) stables around 1 after 60 min. Accordingly, the ethylene FE also gradually increases with Cu(100) formation within the first 35 minutes and then reaches the maximum value. Whether single crystal Cu foils or the time-dependent ethylene selectivity with

structural evolution have verified again that Cu(100) possesses a higher ethylene FE than Cu(111).

Moreover, our Density Functional Theory (DFT) calculations also indicate that Cu(100) has a lower energy barrier for C-C coupling (see Fig.1a in the revised manuscript), which is consistent with our experimental results. Both experimental and DFT results provide solid evidence to confirm that Cu(100) is responsible for the enhancement of ethylene selectivity.

Fig. R12 **a**, OH⁻ electroabsorption profiles on the CuO and CuO-SH electrodes at the potential range of -1 ~ -1.4V, **b**, the surface area ratio of Cu(100) and Cu(111) facets quantified by OH⁻ electroabsorption.

Fig. R13 *In situ* XRD patterns of **a** CuO-SH and **b** CuO collected at various potentials ranging from 0 to -1.4V, **c** Quantitative peak analysis: the ratio of Cu(100) and Cu(111) facets

Fig. R14 **a**, XRD of single crystal Cu(100) and Cu(111), **b,c,d** FE as a function of applied potential over Cu(100) and Cu(111) measured in 0.1M KHCO₃.

Fig. R15. **a**, Time-resolved *operando* XRD patterns CuO-SH electrolysis at -1.4V vs RHE for 120 min in 0.1M KHCO₃, **b**, the corresponding quantitative peak analysis and **c**, CO₂RR

performance of CuO-SH

Our revision: The figure and related discussions have been added in the revised manuscript. Please see the yellow highlighted part on pages 12-14 (lines 283-292, 298-319) in the revised manuscript and pages 21 and 23-24 in the revised supplementary information.

Comment 2 : In Fig. 2c and d, the authors attempt to attribute the higher ethylene FE to a stronger *CO adsorption (characterized by a red-shift of an atop *CO peak). I have a related question and a minor comment. First, how does the authors attribute the stronger *CO adsorption to the thiol ligand? It may as well be just from the increased over-potential. The same data as in Fig. 2d should be provided for the CuO without thiol. Secondly, I suggest the authors to use a brighter color for the peaks in Fig. 2d, the peaks look like dips as it is currently colored.

Our response:

Thank you for your comments. We speculate that the reviewer should refer to Figure 3.

(1) To figure out whether stronger *CO adsorption is attributed to increased over-potential or thiol ligands, we conducted a comparison of the Raman spectra between CuO-SH and CuO under the same potential conditions. It was shown that CuO-SH has a higher peak intensity at the same potential (Fig. R2). By comparing the peak positions of the CO adsorption peaks, we observe that within this potential range, the CO adsorption peak of CuO-SH is slightly lower than that of CuO, indicating a stronger CO binding ability of CuO-SH. Because the low-frequency bonded CO exhibits a stronger binding of carbon to the copper surface. Moreover, the *in situ* infrared spectra show that the CuO-SH exhibits a stronger *CO peak than CuO (see Fig. 5a and 5b in our revised manuscript). Both *in situ* Raman and *in situ* infrared results provide solid evidence to confirm that the DDT modification enhances the CO binding capability of the CuO-SH catalyst.

Indeed, the color of Fig. 3d can easily cause misleading. We have replaced it with Fig. R2 in our revised manuscript.

Our revision: According to the reviewer's constructive comments, the figure and related discussions have been added in the revised manuscript. Please see the yellow highlighted part on pages 10-11(lines 235-238, 242-243) in the revised manuscript.

Comment 3: Along this thread, if the authors would like to design a catalyst with a majority of Cu(100) surface to start with (a Cu cube for example, see *Angew. Chem. Int. Ed.* 2016, 55, 5789 - 5792), and stabilize the surface using the thiol?

Our response:

Thanks for your constructive comment.

(1) According to the reviewer's suggestion, we attempted to synthesize Cu nanocubes following the method reported in [*Angew. Chem. Int. Ed.* **55**, 5789-5792 (2016)]. However, as depicted in Fig. R16, our synthesized Cu nanocubes still exist to a certain extent (111). Unfortunately, after multiple experimental attempts, we didn't obtain perfect Cu(100) nanocube. Thus, we use single crystal Cu(100) foil instead of Cu nanocube. The specific preparation process of single crystal Cu(100) can be found in our responses to comment 1 of reviewer 2.

(2) The Cu(100) single crystal foil was immersed in 5 mL of DDT for a duration of 3 hours. The foil was washed several times with ethanol and dried by compressed N₂. Through *in situ* XRD testing, we discovered that after 30 minutes of CO₂RR reaction at a potential of -1.4 V, the DDT-modified Cu(100) retained its crystal structure, whereas the unmodified Cu(100) underwent partial conversion to Cu(111), as shown in Fig. R17, indicating that DDT is beneficial for stabilizing the Cu(100) during the CO₂RR reaction.

Fig. R16. XRD of prepared Cu100 nanocube.

Fig. R17. XRD of Cu(100) and Cu(100)-SH at -1.4V during CO₂RR in 0.1M KHCO₃.

Our revision: According to the reviewer's insightful comments, the related modification has been made in the revised manuscript. Please see the yellow highlighted part on page 13 (lines 294-296) in the revised manuscript and page 22 in the revised supplementary information.

Comment 4 : Even one is to accept Cu(100)s role in the high FE, whether it is on the catalyst surface by the thiol ligand is a question. The stability test is run only for 5 hours, and during that 5 hour the FE is already decreased from 72% to 60%. To compare, stability in electrochemical ethylene synthesis has been reported in the hundred-hour range for years (for example, Science 360, 783–787 (2018), Nature, 577(7791), 509-513). What leads to a decrease of ethylene FE? Has surface reconstruction taken place? Has the Cu(100) surface been destabilized? Can the authors do some characterization,

an operando XRD for example, to find out? Also, since the authors claim that the key to the high ethylene selectivity is the stabilized Cu(100) as they put into the title, the stability test will be of core importance and it deserves a main-text figure.

Our response:

We appreciate the professional comments. With regard to whether Cu(100) is induced by the thiol ligand, the following evidence can fully illustrate it. Firstly, both *in situ* XRD and OH electroabsorption results of CuO-SH and CuO (Fig. R12 and R13) confirm that the enhanced Cu(100) is attributed to the DDT modification. Secondly, for the single crystal Cu(100) foil as shown in Fig. R17, the thiol ligand greatly promotes the stability of Cu(100). Lastly, our DFT calculations (Fig. 1a in the revised manuscript) also show a stronger binding of the thiol molecule to Cu(100) compared to Cu(111). All the above evidence substantially support that DDT can indeed stabilize the Cu(100) facet.

According to the reviewer's comments on the stability, we conducted a long-term stability test of CuO-SH in a flow cell. The ethylene faradaic efficiency (FE) of CuO-SH started at 73.6% and remained consistent at 49.7% after the 40 hours test under a constant current density of 200 mA/cm² (Fig. R18). Although the FE of ethylene remained at a relatively high level, a gradual decrease in ethylene FE was observed. Therefore, we investigate the structural stability of the CuO-SH catalyst to elucidate the origin of the decrease in ethylene FE. To explore the dynamic structural evolution of the CuO-SH under reaction conditions, time-dependent *in situ* XRD experiments at -1.4 V were performed. We recorded the structural changes of the CuO-SH catalyst at a -1.4V potential over a 2-hour period. In addition, a noticeable decrease of ethylene FE was observed in CuO-SH after 2 hours of stability testing. As shown in Fig. R15, the CuO-SH electrode undergoes gradual reduction to metallic Cu during the first 60 minutes at -1.4 V, and then maintains a relatively stable crystal structure for the remaining 60 minutes.

Furthermore, we doubted that performance decay could be linked to the stability

of surface thiols. We also conducted an investigation on the stability of surface thiols by XPS and XANES. The S2p XPS spectra (Fig. R19a) confirm the partial transformation of DDT on the CuO-SH surface. A new peak attributed to alkanesulfonates appeared at 168 eV in the S2p spectrum of CuO-SH after 5 hours of reaction, and this peak became more pronounced with an extended reaction time of 40 hours. Additionally, the S-K edge X-ray absorption near-edge structure (XANES) spectroscopy of the CuO-SH catalyst revealed an increase of the surface S valence state (indicated by a shift of the edge front peak to higher energies) with increasing reaction time at -1.4 V (Fig. R19b). It can be seen that as the reaction proceeds, the thiol is partially oxidized. We believe that the partial oxidation of thiols will give rise to surface reconstructions. Unfortunately, it is difficult for current techniques to detect such surface reconstruction under *in situ* electrochemical conditions.

In addition, XRD results of CuO-SH after the 40h hour stability test showed that a slight diminishing of the Cu(100) facets of the catalyst occurred due to partial conversion of DDT, as evidenced by a decrease in the Cu(100)/Cu(111) ratio from 1.09 at 2h to 0.87 at 40h (Fig. R20).

According to time-dependent *in situ* XRD, XPS and XANES results, we believe that the decay of ethylene FE of CuO-SH originates from the partial oxidation of the thiols. On the one hand, the transformation of hydrophobic DDT to more hydrophilic alkanesulfonates will facilitate HER process, on the other hand, the oxidation of the thiols induces the decay of Cu(100) facets, thus leading to a decrease of ethylene FE of the CuO-SH in the stability tests.

Fig. R18. The stability of CuO-SH obtained at 200 mA/cm² in a 1 cm² flow cell electrolyser in 1M KOH.

Fig. R19. **a**, XPS spectra and **b**, XANES spectra of CuO-SH before and after electrolysis at -1.4V vs RHE.

Fig. R20 XRD spectra of CuO-SH after electrolysis at 200 mA/cm² for 40 hours in 1M KOH.

Our revision: According to the reviewer's constructive comments, the related modification has been made in the revised manuscript. Please see the yellow highlighted part on page 8 (lines 185-188, 192-197) and page 14 (lines 320-331) in the revised manuscript and pages 12 and pages 24-15 in the revised supplementary information.

Comment 5 : Talking about the performance, I appreciate a high FE, but I am also concerned about the high over-potential where the high FE is achieved. In other words, a high current efficiency may be achieved but at the cost of a low voltage efficiency (the energy efficiency can be thought of as the product of both efficiencies). This can be understood from the operando XRD: the Cu(100) facet is only promoted at a high potential (although I don't understand why). I encourage the authors to consider the energy efficiency, and perform a techno-economic analysis to evaluate the application potential, of their technique.

Our response:

Thanks for the reviewer's positive comments. Indeed, techno-economic analysis will be beneficial in showcasing the application prospects of this work. As regarding why the Cu(100) facet is only promoted at a high potential, we analyze it as follows. As depicted in Fig. R13, the potential-dependent *in situ* XRD of CuO-SH and CuO samples was collected after holding them at each potential for 15 minutes, within the potential range from 0 to -1.4 V. As observed in Fig. R13, Cu(100) in CuO-SH significantly enhanced in the range from -0.8 V to -1.4 V, which is mainly due to the faster reduction rate resulting from the relatively large thermodynamic driving force at high overpotential range. Furthermore, Cu(100) in CuO-SH can be stabilized by the thiol ligand at high overpotential (For details, please refer to our response to the comment 4 raised by reviewer 2.). At low overpotentials, the reduction rate of CuO is relatively slow, and the short reduction time of 15 minutes only allows a partial reduction of CuO to Cu in CuO-SH. However, at high overpotentials, the rate of CuO

reduction increases, leading to a more substantial reduction of CuO to Cu in CuO-SH.

Next, we conducted a techno-economic analysis of the flow cell performance of CuO-SH. The energy conversion efficiency of CuO-SH for ethylene reached 29% at a cell potential of 3.2V in 1M KOH. For the economic analysis, we used the cost of electricity for ethylene production as the index, considering the variability of specific catalysts. According to Table R1, the ethylene power cost of CuO-SH is 1.38 \$/Kg. Comparing the flow cell performance, both the ethylene energy efficiency (EE) and power cost of CuO-SH are superior to some previously reported catalysts. The comparison of EE and production cost suggests that the high FE of ethylene via the strategy of DDT-modified CuO has great potential for future applications.

Table. R1 Comparison of C₂H₄ FE, Energy efficiency, and per electricity cost over CuO-SH with the performances of recently reported catalysts.

	E-cell (V)	FE-C ₂ H ₄ %	Ethylene energy efficiency(%)	Ethylene Electricity cost (\$/Kg)	Reference
0	3.2	79.5	29	1.38	This work
1	2.35	66	32.2	1.25	Nat Catal 3 , 478–487 (2020).
2	2.4	70	34.1	1.17	Science 360 , 783-787 (2018)
3	3.65	65	20.8	1.93	Nature 577 , 509–513 (2020)
4	2.85	68	27.1	1.44	Nat. Catal. 3 , 98–106 (2020)
5	2.7	80	34	1.18	Nature 581 , 178–183 (2020).
6	2.84	72	29	1.36	Nat. Catal. 4 , 20–27 (2021)
7	3.35	60	20.6	1.94	Nat Commun 14 , 2387 (2023).
8	3.33	80	27.6	1.43	Nat Commun 13 , 1877 (2022).

EE is calculated by using the equation: $(1.23 - E_{C_2H_4}) \times FE(C_2H_4) / E_{cell}$, where $E_{C_2H_4} = 0.08$ V, $FE(C_2H_4)$ is the measured Faradaic efficiency for ethylene production in %, and E_{cell} is the cell potential in V, $E_{cell} = E_{anode} - E_{cathode}$. Anode potentials were recorded by a multimeter (Fluke 17B+) with respect to an Ag/AgCl electrode.

Our revision: According to the reviewer's constructive comments, the related modification has been made in the revised manuscript. Please see the yellow

highlighted part on pages 17-18 (lines 410-411) in the revised manuscript and page 33 in the revised supplementary information.

Comment 6: Along this thread: the authors achieve an all-round better performance in a flow cell than in an H-cell (higher FE, much higher current density, lower over-potential). Why didnt the authors present the flow-cell-performance in the main-text and leave the H-cell-performance in the SI?

Our response:

Thanks for the good suggestion. Our initial research objective was to reveal the origin of high ethylene selectivity and the effects of thiol ligands and Cu crystal planes on selectivity and activity, and all *in situ* experiments were conducted in the H-cell so we overlooked the performance of the flow cell. Following the suggestions of the reviewer, we have added the flow cell performance data to Figure 2 in the revised manuscript.

Fig. 2 CO₂RR performances for the CuO and CuO-SH. Faradaic efficiencies (FE) of the CO₂RR products as a function of applied potential over **a** CuO-SH and **b** CuO, **c** FE values for C₂₊ products and FE ratio of C₂₊ products over C₁ products on CuO-SH and CuO at various potentials ranging from -0.8 to -1.6 V, Current densities and product distributions under different potentials over **d** CuO-SH and **e** CuO in a flow cell.

Our revision: According to the reviewer's advice, the related modification has been made in the revised manuscript. Please see the yellow highlighted part on page 9 (lines 206-207) in the revised manuscript.

Comment 7 : In Fig. 3, why do the authors choose different x-domain for the Raman shifts (Panels a, b and d)?

Our response:

Thank you for your careful review. Indeed, the x-domain for these three figures is inconsistent due to our negligence. In the revised version, we have rectified this inconsistency, and the x-axis scale of the three figures has been made consistent.

Fig. 3 *In situ* Raman spectra. *In situ* Raman spectra of **a** CuO-SH and **b** CuO obtained in a potential window OCP to -1.4 V.

Our revision: The related figures have been added in the revised version. Please see the yellow highlighted part on pages 11 (line 239) in the revised manuscript

Comment 8 : Fig. 2, we can see that a major reason for a low ethylene FE for the CuO is the higher HER. What leads to it? Is it again related to the Cu(111) facet? There should be a good explanation.

Our response:

Thanks for your comments. We propose that insufficient mass transfer within the H-cell causes HER to dominate the reaction on the unmodified CuO surface. In the H-cell configuration, the electrode is immersed in a liquid electrolyte

where CO₂ molecules are dissolved. These CO₂ molecules then diffuse downward towards the catalyst surface, driven by a concentration gradient, to participate in the reaction. However, the low solubility and slow diffusion of CO₂ in the electrolyte impose limitations on mass transfer, particularly at high current densities. In comparison, the mass transfer of H₂O and ions is much faster than that of CO₂. Consequently, at high overpotentials or higher currents, a longer diffusion layer thickness of 100 μm can lead to localized under-supply of CO₂ to the catalyst. This localized deficiency of CO₂ availability ultimately results in the HER dominating the reaction instead of the desired CO₂ reduction. Furthermore, the higher HER could be associated with the Cu(111) facets found in unmodified CuO catalysts. Because the CO₂RR intermediate adsorption on Cu suppresses the (HER) due to site blocking effects and/or changes in *H binding energy [*Chem. Rev.* **119**, 7610–7672 (2019)], and Cu(111) has a lower adsorption strength of CO₂*, COOH*, and CO* than Cu(100) [*Nat Catal* **3**, 98–106 (2020)].

Comment 9: Along this thread: Fig. 2 should emphasize more on the C2+:C1 ratio because boosting C-C coupling, instead of HER suppression, is the major theme.

Our response:

Thank you for your meaningful suggestion. In Figure 2, we discussed that the HER suppression is attributed to the surface modification, which regulates the mass transfer of carbon dioxide and protons, thereby suppressing the main competing reactions in carbon dioxide reduction. In response to the reviewer's suggestion, we have moved the C2+/H₂ plot to the Supporting Information and added the C2+/C1 ratio in the main text.

Fig. 2 CO₂RR performances for the CuO and CuO-SH. c FE values for C₂₊ products and FE ratio of C₂₊ products over C₁ products on CuO-SH and CuO at various potentials ranging from -0.8 to -1.6 V.

Our revision: The related figures and discussions have been added in the revised manuscript. Please see the yellow highlighted parts on page 7 (lines 163-170) and page 9 (lines 203-204) in the revised manuscript as well as on page 7 in the revised supplementary information.

Reviewer #3 (Remarks to the Author):

General Comment: In this work, the authors developed a hierarchical nanostructured CuO catalyst stabilized with DDT for CO₂ electrochemical reduction to ethylene. The catalyst was evaluated in both H-cell and flow cell for electrocatalytic performance. The Faradaic Efficiency of C₂H₄ reached up to 72%. Based on the catalytic performance, catalyst characterizations, and theoretical calculations, the authors claimed that DDT treatment of CuO catalyst could stabilize Cu(100) facets and promote the C-C coupling, which leads to high C₂H₄ selectivity and good catalyst stability.

This work aimed to address the great challenges in CO₂RR toward C₂ products and the obtained C₂H₄ selectivity (FE) is among the best reported results in literature. Using thiol to stabilize Cu(100) facets provides a promising route to develop new CO₂RR catalysts toward C₂ products with high selectivities and will benefit the advancement of CO₂RR toward industrial applications. The manuscript was well organized and written. Most of the catalyst characterization, DFT calculation, and catalyst performance evaluations were well designed and support the claims. However, the current work lacks key experiments and characterizations for the CuO-SH stability performance. Therefore, this manuscript is not recommended for acceptance at current version and review is needed after revision.

Our response:

We are grateful to the reviewer for the positive comments on this work, which greatly improve the overall quality of the work. We have considered all suggestions and have modified the manuscript accordingly. The detailed responses to the specific comments are provided as below.

Comment 1: What was the DDT coverage on CuO surface? What was the Thiol/CuO ratio? Any thiol coverage effects on catalytic performance?

Our response:

We appreciate your meaningful comments.

(1) As regarding the surface DDT coverage, we used the method reported in [*Langmuir* **35**, 6888-6897(2019)] to calculate. The surface coverage T(S) of different sulfur species on copper was determined by comparing the peak intensities of the respective component of the S 2p to the Cu 2p 3/2 peak of the substrate as

$$T(S) = \frac{A_S}{A_{Cu}} \frac{S_{Cu}}{S_S} \rho(Cu) \lambda(Cu) \sin\theta \frac{e^{d/[\lambda(S;org)\sin\theta]}}{e^{d/[\lambda(Cu;org)\sin\theta]}} \quad (1)$$

The takeoff angle $\theta = 45^\circ$. The area ratio of the respective S 2p component and Cu 2p 3/2 is denoted as $\frac{A_S}{A_{Cu}}$, where $\frac{S_{Cu}}{S_S}$ represents the ratio of the respective atomic sensitivity factors, $\rho(Cu) \approx 0.14 \times 10^{22}$ atoms/cm³ is the number of copper atoms per unit volume in the surface oxide, $\lambda(Cu) \approx 0.78$ nm is the inelastic mean-free path (IMFP) of Cu photoelectrons in the substrate, $\lambda(S;org)$ and $\lambda(Cu;org)$ are the IMPFs of the respective photoelectrons in the organic layer, and d represents the thickness of the DDT. For sulfur and copper photoelectrons, the IMPFs of S 2p and Cu 2p 3/2 in the organic monolayers are $\lambda(S; org) \approx 3.81$ nm and $\lambda(Cu; org) \approx 2.12$ nm. $d \approx 10$ nm according to TEM. We calculated the coverage of CuO-SH to be 1.06 molecules/ nm², based on Eq. 1 and the results of Fig. R21a-b, and the atomic ratio of S: Cu in the ratio of thiols to CuO can be roughly determined as 5.94: 6.37 based on XPS analysis.

To clarify the effect of different DDT coverage on the catalytic performance of CuO-SH, we prepared CuO-SH-1 and CuO-SH-3 samples by shortening the thiol treatment time to 10 min or extending it to 30 min, respectively. Similarly, we also calculated the DDT surface coverage of CuO-SH-1/-3 by Eq. (1), as shown in Table R2, as 0.59 molecules/ nm² and 1.31 molecules/ nm², respectively. We also verified the enhancement of the surface DDT with the time of thiol treatment by ATR-IR, because the intensity of the IR C-H peak of the samples was enhanced with the time of surface treatment. We then evaluated the catalytic performance of the three samples in the H-cell with 0.1 M KHCO₃ as the electrolyte. From Fig. R21d, it can be seen that enhancing the DDT coverage of CuO-SH is beneficial to promote the generation of C₂H₄, but

further enhancement of the coverage rather leads to a decrease in the performance, meaning the existence of an optimal thiol coverage.

Fig. R21. The Cu2p (a), S2p (b) spectra and (c) FT-IR spectra of CuO-SH catalysts after different treatment time, and the corresponding C₂H₄ faradic efficiency in 0.1M KHCO₃.

Table. R2. Element analysis and DDT coverage.

Sample	Atomic %				DDT-Coverage (molecules/ nm ²)
	C	O	Cu	S	
CuO-SH-1	75.38	13.02	6.66	4.94	0.59
CuO-SH-2	78.93	8.75	6.37	5.94	1.06
CuO-SH-3	82.26	4.07	6.97	6.7	1.31

Our revision: The corresponding discussion and figures have been added. Please see the yellow highlighted parts on page 35-36 in the Supplementary Information.

Comment 2: Based on 5 hours stability tests, it seems that CuO-SH showed a significant performance degradation with a descending trend, although the rate,

as claimed by authors, was slower (16%) than that of CuO (48%) in 5 hours. Any interpretation of what causes such performance degradation of CuO-SH?

Our response:

Thank you for your comments.

We investigate the structural stability of the CuO-SH catalyst to elucidate the origin of the decrease in ethylene FE. To explore the dynamic structural evolution of the CuO-SH under reaction conditions, time-dependent *in situ* XRD experiments at -1.4 V were performed. We recorded the structural changes of the CuO-SH catalyst at a -1.4V potential over a 2-hour period. In addition, a noticeable decrease of ethylene FE was observed in CuO-SH after 2 hours of stability testing. As shown in Fig. R15, the CuO-SH electrode undergoes gradual reduction to metallic Cu during the first 60 minutes at -1.4 V, and then maintains a relatively stable crystal structure for the remaining 60 minutes.

Furthermore, we doubted that performance decay could be linked to the stability of surface thiols. We also conducted an investigation on the stability of surface thiols by XPS and XANES. The S2p XPS spectra (Fig. R19a) confirm the partial transformation of DDT on the CuO-SH surface. A new peak attributed to alkanesulfonates appeared at 168 eV in the S2p spectrum of CuO-SH after 5 hours of reaction, and this peak became more pronounced with an extended reaction time of 40 hours. Additionally, the S-K edge X-ray absorption near-edge structure (XANES) spectroscopy of the CuO-SH catalyst revealed an increase of the surface S valence state (indicated by a shift of the edge front peak to higher energies) with increasing reaction time at -1.4 V (Fig. R19b). It can be seen that as the reaction proceeds, the thiol is partially oxidized. We believe that the partial oxidation of thiols will give rise to surface reconstructions. Unfortunately, it is difficult for current techniques to detect such surface reconstruction under *in situ* electrochemical conditions.

In addition, XRD results of CuO-SH after the 40h hour stability test showed that a slight diminishing of the Cu(100) facets of the catalyst occurred due to partial conversion of DDT, as evidenced by a decrease in the Cu(100)/Cu(111)

ratio from 1.09 at 2h to 0.87 at 40h (Fig. R20).

According to time-dependent *in situ* XRD, XPS and XANES results, we believe that the decay of ethylene FE of CuO-SH originates from the partial oxidation of the thiols. On the one hand, the transformation of hydrophobic DDT to more hydrophilic alkanesulfonates will facilitate HER process, on the other hand, the oxidation of the thiols induces the decay of Cu(100) facets, thus leading to a decrease of ethylene FE of the CuO-SH in the stability tests.

Our revision: The related modification has been made in the revised manuscript. Please see the yellow highlighted part on page 8 (lines 192-197) and page 14 (lines 320-331) in the revised manuscript and page 12 and pages 24-25 in the revised supplementary information.

Comment 3: Did authors observe any structure changes of CuO-SH after stability testing? XRD results after stability testing?

Our response:

Thank you for this comment. XRD results of CuO-SH after 40h hour stability test showed a slight diminishing of the Cu(100) facets of the catalyst as evidenced by a decrease in the Cu(100)/Cu(111) ratio from 1.09 at 2h to 0.87 at 40h(Fig. R20). The reduction of Cu(100) facets on CuO-SH during the test could be the result of partial transformation of the DDT (Fig. R19).

Comment 4: It seemed the authors did stability testing in an H-cell at a constant voltage for 5 hours. Usually, a 5-hour test is too short for an electrode catalyst stability evaluation. Since the authors also checked CuO-SH performance in a flow cell, it will be highly desirable to run the stability testing in the flow cell under constant current operation mode (200 or 300 mA cm⁻² current density) for a longer time, at least several tens of hours. Post characterizations of thiol, and possible structure changes should be conducted after such long-term testing. This will be critical to evaluate the potential application of the proposed thiol-catalyst system regarding the industrial application requirements.

Our response:

Thank you for the great comments. According to the reviewer's suggestion, we have conducted a long-term stability test of CuO-SH in a flow cell. During the long-term stability test lasting 40 hours, the ethylene faradaic efficiency (FE) of CuO-SH started at 73.6% and remained consistent at 49.7% under a constant current density of 200 mA/cm² and FE of H₂ increased from 4.82% to 16.83% during the 40-hour reaction (Fig. R18). The S-K edge XANES spectroscopy in conjunction with the S2p spectrum further confirms the partial transformation of DDT on the CuO-SH surface (Fig. R19). XRD results of CuO-SH after the 40h hour stability test showed that a slight diminishing of the Cu(100) facets of the catalyst occurred due to partial conversion of DDT, as evidenced by a decrease in the Cu(100)/Cu(111) ratio from 1.09 at 2h to 0.87 (Fig. R20). The transformation of hydrophobic DDT to more hydrophilic alkanesulfonates facilitating HER, and the oxidation of the thiols induced the decay of Cu(100) facets are responsible for the decrease of ethylene FE of the CuO-SH catalysts in the stability tests. It is crucial to enhance stability for future industrial applications, which still need further studies.

Our revision: The related modification has been made in the revised manuscript. Please see the yellow highlighted part on page 8 (lines 185-188) and page 14 (lines 320-331) in the revised manuscript and pages 12 and pages 24-25 in the revised supplementary information.

Comment 5: For the ECSA measurements, did the authors do the IR correction? The potential scan range (0.3 V) was much wider than the normal one (0.1V) used for such measurements. I suggest running the potential scan between 0.175 V and 0.275V.

Our response:

Thank you for the technical comment. We performed ECSA testing in the range of 0.175 V to 0.275 V, which was similar to the results obtained in the potential scan range of 0.3 V (Fig. R22). Although larger scan windows (0.2V or 0.3V)

have been used in some other reports, it is noteworthy that the more common and widely adopted choice, as recommended by the reviewers, remains the use of a 0.1V scan potential interval for ECSA (electrochemical surface area) testing of catalysts [*Chem. Soc. Rev.*, **48**, 2518--2534(2019)]. Therefore, we replaced the original data with values from a narrower range.

Fig. R22. ECSA measurement of CuO-SH and CuO: CV curves for CuO-SH (a) and CuO (b) obtained in capacitance region at varying scan rates. (c) Capacitance current density at 0.257 V vs. RHE as a function of scan rate. (d) Comparison of ECSA measure at different potential scan regions.

Our revision: According to the reviewer's suggestions, the corresponding revisions have been made. Please see the yellow highlighted part yellow highlighted parts in page 11 (lines 250-253, 255) in the revised manuscript as well as page 16 in the revised supplementary information.

Comment 6: Line 379 Nafion membrane (N-115, dubant), please clarify

Our response: We thank the reviewer for the careful review. The “dubant” was have been revised to “DuPont” in the revised version.

Our revision: The typo has been revised. Please see the yellow highlighted part yellow highlighted parts in page 19 (line 441) in the revised manuscript

Comment 7: Line 382-383 working electrode area was 1 cm^2 (0.5 cm x1cm x2) please clarify part in the brackets.

Our response: Thanks for your comments. It is indeed easy to cause misunderstandings. 2 in “(0.5 cm × 1 cm × 2)” means two sides of the electrode. To avoid misleading, it has been changed to (0.5 cm × 1 cm × 2 side).

Our revision: The corresponding revisions have been made. Please see the yellow highlighted part yellow highlighted parts in page 19 (line 448) in the revised manuscript

Comment 8: Line 388-395, please clarify those in the brackets (400 uL) of well-mixed catalyst ink containing (20 mL) of Nafion dispersion, please also clarify how the 1 M KOH electrolyte circulated during the flow cell test.

Our response:

We thank for your comments. We are sorry for this typo, and “(20 mL) of Nafion dispersion” has been revised to“(20 μL) of Nafion dispersion” in the revised manuscript. Moreover, the circulation of the electrolyte is achieved through the green and red pipelines shown in Figure R34, during the flow cell test.

Fig. R23. Model of flow cell system.

Our revision: The typo has been revised. Please see the yellow highlighted part yellow highlighted parts in page 20 (line 455) in the revised manuscript

Comment 9: Line 391, was used for flow cell testing, in the caption of Supplementary Fig. 8, 0.1 M KHCO_3 electrolyte, please clarify.

Our response:

Thanks very much for the reviewer's careful check. We are sorry for this mistake. "0.1 M KHCO_3 " has been revised to "1 M KOH " and we have added the flow cell performance data into Fig. 2 in the revised manuscript.

Our revision: The corresponding revisions have been made. Please see the yellow highlighted part yellow highlighted parts in page 9 (lines 204-206) in the revised manuscript

Comment 10: Line 415-417, this may cause confusion since the sentence is for working electrode treatment while the supplementary Fig.17 is the photo of the *in situ* ATR-FTIR setup.

Our response:

Thank you for the reviewer's careful reading and kind reminding. The following words "The *in situ* ATR-FTIR was performed on a Bruker INVENIO spectrometer with a HgCdTe(MCT) detector cooled with liquid nitrogen. The measurement was conducted in a homemade electrochemical H-cell furnished

with Pt-wire and Ag/AgCl as counter and reference electrodes. A fixed-angle Si prism (60°) coated with catalysts embed into the bottom of the cell served as the working electrode. All spectra were collected with 128 scans and a resolution of 4 cm⁻¹ (Supplementary Fig. 30).” have been added in the revised manuscript.

Our revision: The *in situ* ATR FTIR experiment details have been added. Please see the yellow highlighted part yellow highlighted parts in page 21 (lines 483-488) in the revised manuscript.

Reviewers' comments:

Reviewer #1 (Remarks to the Author):

The authors have done a great job assimilating information from the reviewers and incorporating additional experiments and data to their revised manuscript. Their results do show conclusive evidence of CuO-SH catalysts having better ethylene selectivity than CuO for the CO₂RR, but their analysis of reasons to why that is the case can be strengthened by addressing the following comments:-

Major Comments

- 1) Fig 2f shows that the ethylene FE of CuO-SH started at 73.6% and dropped to 49.7% after 40 hrs under a constant current density. By just comparing numbers of 49.7% ethylene FE here to the 44.8% FE obtained with CuO at -1.2 V, I am inclined to hypothesize that under long term operation conditions, the CuO-SH does reconstruct to give similar performance to CuO. The authors do perform XAFS after the 40 h reaction time and notice a new peak attributed to alkanesulfonates. Is this peak responsible for the decreasing ethylene FE? If not, do the authors have a hypothesis of why the FE drops off in spite of most of the DDT molecules remaining (Supplementary Fig 7.)?

I noticed that the last paragraph on page 14 before the mechanistic insight mentions a potential reason for this loss in ethylene FE over time. I would recommend moving that paragraph closer to the discussion around Fig 2f. If not at least a mention during the discussion around Fig 2f that there is a paragraph later that can explain the loss in FE over time would suffice.

- 2) Attributing the increased linearly bonded CO coverage in CuO-SH vs. CuO to Cu(100) without experimental proof on page 15 line 353-355 is difficult to believe given there is literature showing that ATR-FTIR can detect different peak positions for adsorbed CO on different facets. This work ([https://doi.org/10.1016/S0013-4686\(98\)00261-8](https://doi.org/10.1016/S0013-4686(98)00261-8)) shows how peak positions on Cu(100) are different from polycrystalline Cu. Similarly, Hollins and Pritchard have extensive work discussing CO peak positions on different facets. Supplementary Figure 23. only shows DFT results for Cu(100), how does the adsorption energy compare to say Cu(111) or other facets? Why is it inferred that Cu(100) is the most favorable facet to bind to without these results?
- 3) The authors have shown Stark tuning for the adsorbed COOH impurity which is great. I would recommend including this in the SI and not leave it as a for review only figure. I would also add the wavenumbers over which the Stark tuning occurs (like peak position at say the lowest and highest potential) to give readers a sense of how much the peak moves with potential. How about for the case for the adsorbed CHO and adsorbed CO-CHO impurities, do those show Stark tuning too to confirm they are surface adsorbed?

Minor comments

- 1) On line 118 at the end of page 4, the authors mention that hydrophobic salt layer being introduced on the CuO surface creates "a three-phase interface and thereby enhancing local

CO₂ concentration.” It would be clearer for readers if the authors elaborate on what the three phases are and even more importantly why this configuration enhances the local CO₂ concentration?

- 2) On line 176 at the end of page 7, the authors mention that “the decay ratios of the CuO-SH and CuO are 16% and 48%.” While that is true it would be better to specify that those decay ratios are from the max FE and not from the initial FE. From the initial FE the CuO case only decays for 21 to 18% which is not a 48% decrease.
- 3) In supplementary Fig 9., either in the figure caption or in the main text, the authors need to specify that the higher peak area/intensity of CO₂ (aq) for the CuO-SH case compared to the CuO case signifies that the concentration of CO₂ near the interface is higher. This will help folks not familiar with reading iR graphs understand how they reach that conclusion. Since, those two traces are collected on different substrates, comparing the peak intensity directly is not ideal. A more rigorous approach would be to compare the CO₂ (aq) peak intensity to a reference peak for both cases. This way, concerns about seeing a low peak intensity because of maybe a thicker CuO substrate than CuO-SH leading to a lower signal can be quelled. The authors used a similar approach when comparing Raman peak intensities by using the glassy carbon peak as a reference. I would also add scale bars for the intensity to all spectroscopic plots to again quell similar concerns that one trace might have been scaled to make peaks look more intense.
- 4) The authors claim in Fig 5a and b that the peak intensity of linearly bonded CO is higher for the CuO-SH case than the CuO one. I would recommend using a reference peak to quell concerns about substrate thickness leading to differences in signal like mentioned in the previous comment.

Reviewer #2 (Remarks to the Author):

Y. Yao et al. has resubmitted the manuscript with a substantial supplementary data, with a good amount of effort on the verification of the Cu(100) surface’s role in CO₂RR. I should admit that I am impressed by the amount of work conducted in the revision round, which really clarify many of my questions. However, I should also say that a part of the data may act as a “double-edged sword” as it also clarifies some of my concerns, especially related to the novelty of the work.

- It is an previously accepted concept that Cu(100) promotes C-C coupling and thereby promoting C₂H₄ production. So the novelty of the present work should be on the STABILIZATION of the Cu(100) facet, which is also what the title implies. However, although the authors have supplemented a time-lapse test, it is only for 40 hours, during which the C₂H₄ FE drops from 70% to 50%. The authors suspect that the surface thiol is unstable. Will this catalyst be practical?

- The authors have performed CO₂RR experiments on single crystal Cu(100) and Cu(111) surfaces, and attempted to verify the Cu(100)’s role on C₂H₄ production. While a differential performance was shown,

but why the C₂H₄ FE is so much lower? Even one is to ignore the high HER for the moment, a high C₂H₄ selectivity among all carbon-based products should be expected, at least before the single crystal Cu(100) degrades (if it does degrade). But in the reality even more C₁ products (CH₄ + CO) is produced than C₂. I feel that this is not only the matter of comparing Cu(100) and Cu(111). Clearly here facet is not the sole control factor (maybe not even the most important one), because if it is, then the single crystal Cu(100) should have the best C₂H₄ selectivity.

- I admit that the work reports a high C₂H₄ FE – about the same as the highest previously reported value (in 2020). However, other performance factors are modest: a high over potential results in a modest energy efficiency, and a stability of only 40 hours.

- To summarize: I am trying to think of a good novelty that can guarantee publication in Nature Communications. At the moment there are two points of potential novelty I can see: 1) a DDT can promote a Cu(100) surface during the electroreduction of CuO, and 2) a ~80% C₂H₄ FE that ties with previous records (at the cost of higher overpotential). I am not sure if the two points of novelty merits publication in Nature Communications. With that said, I am interested to see how the authors may defend the novelty of their work, I am happy to do another round of review if needed.

Reviewer #3 (Remarks to the Author):

In this revised manuscript, the authors made thorough responses to all the comments from the reviewers. Additional experiments were conducted for some of the comments. The experiments were well designed and the obtained results provide solid evidence to answer the questions raised by the reviews and well support the authors' claims. The authors also made corresponding changes and corrections based on the reviewers' comments. The revised manuscript provides a promising route to develop new CO₂RR catalysts toward C₂ products with high selectivities by using thiol (DDT) to stabilize the Cu (100) facet. This work will benefit the understanding of CO₂RR catalysts design toward C₂ products and the mechanism of CO₂RR to C₂ products. The revised manuscript was well organized and written. The experiment results and data analysis support the claims. Therefore, the revised manuscript is recommended for acceptance after addressing the following comments:

1)For the flow cell experiment, were the anolyte and catholyte circulated? In Line 451, it was said "Two parallel fluxes 1 M KOH were injected". Based on Fig. R23, it seems the electrolyte was circulated. Was a reference electrode used?

2)The authors used several pale colors for plots, labels and titles in the figures (yellow, for instance), which is not easy to see for the readers (fig1e, fig 4a, b, fig 5 a, b, c). It will be much better if a dark color is used for those.

3)Please check the grammar and format of the manuscript.

Point-by-point responses to Reviewers' Comments

Reviewer #1 (Remarks to the Author):

The authors have done a great job assimilating information from the reviewers and incorporating additional experiments and data to their revised manuscript. Their results do show conclusive evidence of CuO-SH catalysts having better ethylene selectivity than CuO for the CO₂RR, but their analysis of reasons to why that is the case can be strengthened by addressing the following comments:

Our response: We thank the reviewer for the positive comments, and appreciate the thoughtful and constructive suggestions to improve the overall quality of the work. All these insightful suggestions raised by the reviewer have been thoroughly considered and the corresponding revisions have been made as below.

Major Comments

1) Fig 2f shows that the ethylene FE of CuO-SH started at 73.6% and dropped to 49.7% after 40 hrs under a constant current density. By just comparing numbers of 49.7% ethylene FE here to the 44.8% FE obtained with CuO at -1.2 V, I am inclined to hypothesize that under long term operation conditions, the CuO-SH does reconstruct to give similar performance to CuO. The authors do perform XAFS after the 40 h reaction time and notice a new peak attributed to alkanesulfonates. Is this peak responsible for the decreasing ethylene FE? If not, do the authors have a hypothesis of why the FE drops off in spite of most of the DDT molecules remaining (Supplementary Fig 7.)? I noticed that the last paragraph on page 14 before the mechanistic insight mentions a potential reason for this loss in ethylene FE over time. I would recommend moving that paragraph closer to the discussion around Fig 2f. If not at least a mention during the discussion around Fig 2f that there is a paragraph later that can explain the loss in FE over time would suffice.

Response : Thank you for your comments. (1) Reviewers assumed that the restructuring of the CuO-SH catalyst leads to a similar performance comparable

to CuO. From the XRD results of CuO-SH (after 40 hours of testing) and CuO (fresh sample) in Figure R1g, we can see that there are significant differences in their structures, i.e. the intensity of Cu(100) of CuO-SH is much higher than that of CuO. To further clarify this issue, we conducted a stability testing of CuO and a comparative XRD analysis of CuO-SH and CuO after testing under identical conditions (constant current of 200 mA/cm² in a 1M KOH flow cell). The performance of CuO deteriorated rapidly over the course of 25 hours of continuous operation, with the ethylene FE dropping to less than 10% (Fig. R1a). The significant degradation of ethylene FE observed in the CuO catalyst can be attributed to the structural transformation of the CuO electrode during the test. As shown in Figure R1b-c, the CuO nanowires underwent a dramatic surface change (transformed into broken surfaces), and its ECSA significantly reduced by 64% after 40 hours of testing (Fig. R1f). In contrast, the CuO-SH electrode maintained its one-dimensional nanowire morphology, bulk crystal structure and ECSA throughout the test period (Fig. R1d-e), highlighting the improved stability of the modified CuO-SH electrode. As for whether its surface structure has been reconstructed, to be honest, we currently lack understanding of atomic level surface reconstruction during stability testing due to limitations in characterization techniques. Nonetheless, there are significant differences in the bulk crystal structure.

(2) As the reviewer mentioned, we consider that the performance degradation of CuO-SH may be primarily related to the partial loss of thiol groups (oxidation or detachment). Further analysis of the S2p spectrum of CuO-SH revealed that, although most of the DDT remained intact, there was still a 17.3% DDT loss as shown in Fig.R2a. To reduce experimental errors, we conducted repetitive experiments. We collected S2p spectra from two different CuO-SH samples after testing and compared them with the S2p spectrum before testing (Fig.R2b-c). It was observed that after 40 hours of testing, the DDT loss ranged from 16% to 21%. In addition to the partial conversion of DDT into alkanesulfonates, we

speculate that some DDT may have also detached, as we detected the presence of DDT in the electrolyte (Fig.R3). After 40 hours of testing the CuO-SH electrode, we collected the H-NMR spectrum of the upper electrolyte and the C-H signal (-CH₂- in DDT or alkanesulfonates) was detected (see the H-NMR spectrum of DDT for comparison), indicating that some of the DDT or alkanesulfonates had leached from the electrode during the reaction. After 40 hours of operation with the CuO-SH catalyst, a noticeable decrease in ethylene FE was observed (from 73.6% to 49.7%), while DDT showed a decrease of 17.3% (Fig.R4). In addition, a considerable increase in selectivity towards C1 products (from 5.6% to 17.2%) and H₂ (from 4.8% to 16.9%) was observed (Fig.R4). The loss of surface-bound thiol groups can lead to a decrease in the selectivity of ethylene, and the conversion of surface thiols to hydrophilic sulfonic acid groups can enhance the HER. Therefore, we deduce that the reduction in thiol groups is the primary cause of the stability decline observed in CuO-SH catalyst.

To further verify the effect of thiols on stability , we used thiols to retreat the CuO-SH samples after stability testing. As shown in Figure R5, re-treatment of the CuO-SH electrode with DDT (immersing the electrode in a DDT solution for 5 minutes, followed by ethanol rinsing and nitrogen drying) resulted in a 27.5% increase in surface thiols recovery. We then performed further CO₂RR performance tests on the re-treated samples and observed an improvement in the FE of ethylene (from 46.3 to 64.1%). This finding not only confirms the important impact of thiols on the selectivity of ethylene and stability but also provides a feasible path to recover its stability.

(3) According to the reviewer's suggestion, we have added the discussion section regarding the decline in ethylene FE on page 8, immediately following the content related to the stability results.

Our revision: According to the reviewer's insightful comment, the related discussion and figures have been added in the revised manuscript and the

revised supplementary information. Please see the yellow highlighted parts on page 8 (lines 185-187, 196-198, 200-207) in the revised manuscript, and pages 11-12, pages 15-16 in the revised supplementary information.

Figure.R1, **a** Stability test for CuO-SH and CuO at the current density 200 mA cm^{-2} in a flow cell, SEM images of CuO(**b**) and CuO-40h (**c**), CuO-SH (**d**) and CuO-SH-40h (**e**) and their ECSA (**f**) comparison, **g**, The XRD of CuO-2h CuO-40h and CuO-SH-40h. The CuO and CuO-SH represent the catalysts as prepared and the CuO-2h, CuO-40h and CuO-SH-40h represent the catalysts after 2 and 40 hours of testing respectively.

Figure.R2, The S2p of CuO-SH after stability test (a) and the S2p results of two different CuO-SH samples after testing in repetitive experiments (b-c).

Figure.R3, a, The 1H-NMR of the upper electrolyte solution after 40 hours of constant current electrolysis at 200 mA/cm² and b, A reference spectrum of dodecanethiol in CDCl₃

Figure.R4, The products FE comparison of the CuO-SH at 0.5 and 40 hours of constant current electrolysis at 200 mA/cm².

Figure.R5, The S₂p spectrum (left) and products FE comparison (right) of the CuO-SH at 40 hours and after DDT-Retreatment.

2) Attributing the increased linearly bonded CO coverage in CuO-SH vs. CuO to Cu(100) without experimental proof on page 15 line 353-355 is difficult to believe given there is literature showing that ATR-FTIR can detect different peak positions for adsorbed CO on different facets. This work ([https://doi.org/10.1016/S0013-4686\(98\)00261-8](https://doi.org/10.1016/S0013-4686(98)00261-8)) shows how peak positions on Cu(100) are different from polycrystalline Cu. Similarly, Hollins and Pritchard have extensive work discussing CO peak positions on different facets.

Supplementary Figure 23. only shows DFT results for Cu(100), how does the adsorption energy compare to say Cu(111) or other facets? Why is it inferred that Cu(100) is the most favorable facet to bind to without these results?

Response : Thanks for your insightful comments. The reviewer raised concerns regarding the lack of experimental evidence supporting the contribution of Cu(100) facets to the linearly bonded CO (CO_L) coverage and the absence of an adsorption energy comparison with Cu(111).

(1) In order to address the first issue, we collected ATR-IR spectra of Cu(100) foils, Cu(111) foils, CuO and CuO-SH in a CO-saturated 0.1M KHCO_3 solution. As displayed in Figure R6a-c, the vibrational bands centered at ca. 2040 cm^{-1} , were observed over the entire potential range (-0.2 to -1 V vs. RHE), which can be attributed to the linear adsorption of CO. The peak position of the CO adsorption peak on Cu(100) foil ($2042\text{-}2014\text{ cm}^{-1}$) showed a more positive shift compared to Cu(111) foil ($2031\text{-}2006\text{ cm}^{-1}$), consistent with previous findings (Table.R1). Moreover, the intensity of the CO adsorption peak on the surface of the Cu(100) foil is stronger than that observed on Cu(111). These findings suggest that the Cu(100) facet could be a stronger $\ast\text{CO}$ adsorption site compared to Cu(111). In addition, the peaks of adsorbed CO_L over CuO-SH are in higher wavenumbers than CuO, corresponding to the strong binding of CO_L on CuO-SH with a higher exposure of Cu(100) facets. The ATR-IR results confirm that a higher exposure of Cu(100) facets is beneficial to the increase of strong adsorption sites on the surface that maintains a higher $\ast\text{CO}$ coverage on the surface.

(2) Furthermore, we have also included additional calculations for the linear CO adsorption energies on the Cu(111) within the investigated range of CO coverage, as shown in Figure R6d. The results again consistently show that CO exhibits stronger binding to the Cu(100) surface compared to Cu(111).

Therefore, based on the ATR-IR data and comprehensive adsorption energy calculations on Cu(100) and Cu(111) surfaces, it can be concluded that

increasing the exposure of Cu(100) facets is able to enhance CO_L coverage on CuO-SH surfaces.

Our revision: According to the reviewer's constructive comments, the figure and the related discussions have been added in the revised manuscript and the revised supplementary information. Please see the yellow highlighted part on page 15 (lines 362-363) in the revised manuscript and pages 33-34 in the revised supplementary information.

Figure.R6, the ATR-IR spectrum of **a**, Cu(100) and Cu(111) foil, **b**, CuO-SH, **c**, CuO collected in CO saturated 0.1M KHCO₃, **d**, the adsorption energy comparison between Cu(100) and Cu(111).

Table.R1 The summary of infrared adsorption peak positions under different conditions for CO.

	Facet	Wavenumber (cm ⁻¹)	E (V vs SHE)	Condition	Ref
1	Cu(111)	2070-2078	-	UHV	Surface Sci. 89 486-495 (1979).
2	Cu(100)	2079-2087	-	UHV	Surface Sci. 55 701 (1976).
3	Cu(100)	2020-2060	-0.4~-0.9	0.1M KClO ₄	ACS Catal. 11 , 3173-3181(2021)
4	Cu(100)	2050	-0.88	CO sat PBS 1.6-2.0°C	Electrochim Acta 44 1389-1395 (1998)
5	Poly-Cu	2078	-0.4	CO sat PBS 1.6-2.0°C	Electrochim Acta 44 1389-1395 (1998)

3) The authors have shown Stark tuning for the adsorbed COOH impurity which is great. I would recommend including this in the SI and not leave it as a for review only figure.

I would also add the wavenumbers over which the Stark tuning occurs (like peak position at say the lowest and highest potential) to give readers a sense of how much the peak moves with potential. How about for the case for the adsorbed CHO and adsorbed CO-CHO impurities, do those show Stark tuning too to confirm they are surface adsorbed?

Response :

Thank you for your comment. Following the reviewer's suggestions, we have added Table R2 in the Supplementary Information along with corresponding figures to provide a more intuitive understanding of the peak shifts. As shown in Fig.R6, the Stark tuning of the CHO and CO-CHO intermediates is observed as the potential becomes more negative, confirming their adsorption as intermediates on the catalyst surface.

Our revision: The figures and the related discussions have been added in the revised manuscript and the revised supplementary information. Please see the yellow highlighted part on page 15 (lines 356-358) in the revised manuscript and page 32 in the revised supplementary information.

Table.R2 Peak position of intermediates detected in IR experiment.

	Intermediates species	Potential(V vs RHE)	wavenumber(cm^{-1})
CuO-SH	~COOH	-0.2 ~-1.4	1401-1388
	~CO-CHO	-0.2 ~-1.4	1596-1582
	~CHO	-0.4 ~-1.4	1723-1710
	~CO	-0.2~-1.4	2043-2012
CuO	~COOH	-0.1 ~-1.4	1409-1381
	~CO	-0.2~-1.4	2039-2011

Fig.R7 *In situ* ATR-FTIR recorded at different applied potentials for a CuO-SH and b CuO catalysts.

Minor comments

1) On line 118 at the end of page 4, the authors mention that hydrophobic salt layer being introduced on the CuO surface creates “a three-phase interface and thereby enhancing local CO₂ concentration.” It would be clearer for readers if the authors elaborate on what the three phases are and even more importantly why this configuration enhances the local CO₂ concentration?

Response : Thanks for your comments. As shown in Fig. R8, the three phases between the CuO-SH nanowires, electrolyte, and CO₂ were created by the super-hydrophobicity of CuO-SH. The surface modified-DDT layer leads to the limitation of liquid H₂O transportation which could effectively reduce the liquid volume fraction around Cu nanowire structures to form an electrolyte-solid-gas three-phase interface. On the one hand, the interface could promote the mass transfer of CO₂ and significantly enhance the local CO₂ concentration surrounding the catalyst surface[(*J. Am. Chem. Soc.* **145**, 11323–11332 (2023))]. On the other hand, the structures resembling spider hydrophobic hairs form by DDT-modified CuO nanowires, exhibiting hydrophobicity at both micro and nanoscales, which can lead to the phenomenon of gas entrapment, where air pockets are captured, as reported in "*Nat. Chem.* **10**, 974-980 (2018)." Therefore, the enhanced local CO₂ concentration was achieved in the CuO-SH as illustrated by our ATR-IR experiment. (Supplementary Fig. 9).

Our revision: The figure and the related discussions have been added in the revised supplementary information. Please see the yellow highlighted part on page 18 in the revised supplementary information.

Fig. R8. The illustrations of the operation of the hydrophobic CuO nanowires show the enhanced CO₂ mass transport from the triple-phase boundary between the electrolyte, electrode

2) On line 176 at the end of page 7, the authors mention that “the decay ratios of the CuO-SH and CuO are 16% and 48%.” While that is true it would be better to specify that those decay ratios

Response : Thank you for your comments. To avoid any ambiguity, we have provided a more explicit description of the decay ratios. In the revised manuscript, we have replaced the sentence "The decay ratios of CuO-SH and CuO are 16.7% and 48.6%, respectively" with "The C₂H₄ FE decay ratio of CuO-SH and CuO are 16.7% (from 72% to 60%) and 48.6% (from 35% to 18%), respectively, after 5 hours of operation".

Our revision: The related modifications have been made and we have moved the discussions regarding the 5-hour stability test to the revised supplementary information, which are replaced by the discussions for the 40-hour stability test of CuO-SH and CuO in a flow cell. Please see the yellow highlighted part on page 12 in the revised supplementary information.

Reviewer #2 (Remarks to the Author):

Y. Yao et al. has resubmitted the manuscript with a substantial supplementary data, with a good amount of effort on the verification of the Cu(100) surface's role in CO₂RR. I should admit that I am impressed by the amount of work conducted in the revision round, which really clarify many of my questions. However, I should also say that a part of the data may act as a “double-edged sword” as it also clarifies some of my concerns, especially related to the novelty of the work.

Our response:

We are grateful to the reviewer for the positive comments on this revision. As for the remaining concerns of the reviewers, the detailed responses are provided as below.

1) It is an previously accepted concept that Cu(100) promotes C-C coupling and thereby promoting C₂H₄ production. So the novelty of the present work should be on the STABILIZATION of the Cu(100) facet, which is also what the title implies. However, although the authors have supplemented a time-lapse test, it is only for 40 hours, during which the C₂H₄ FE drops from 70% to 50%. The authors suspect that the surface thiol is unstable. Will this catalyst be practical?

Response : Thank you for your comments. Indeed, the stability of catalysts is crucial for future applications as pointed out by the reviewer. Considering the loss of surface thiol groups leading to the decline of stability, we further performed a surface re-treatment by dropping 20 μL of 20 mM DDT-ethanol solution onto the gas diffusion electrode (GDE) and washing the CuO-SH catalyst with ethanol. The Re-treated CuO-SH catalyst showed an improved ethylene FE of 63% as shown in Figure R8. Following two rounds of re-treatment, the ethylene Faradaic efficiency consistently maintained a relatively high level (52.1%) after 160 hours of operation, thus demonstrating the enhanced stability of the CuO-SH catalyst. Nevertheless, there is still a long

way to go in practical applications, and lots of problems need to be fixed, including selectivity, activity, yield, stability and cost etc. All issues need to be solved step by step. In this work, we provide a technical path to improve the selectivity of ethylene products and try our best to reveal the underlying reasons for the high selectivity via various in-situ techniques. As for stability, further research and improvement are needed, but it's difficult to solve all the problems in one paper.

Our revision: According to the reviewer's constructive comments, the figure and the related discussions have been added in the revised manuscript. Please see the yellow highlighted part on page 8 (lines 205-207) in the revised manuscript.

Figure. R9, Stability test for CuO-SH at 200 mA / cm² current density in a flow cell.

2) The authors have performed CO₂RR experiments on single crystal Cu(100) and Cu(111) surfaces, and attempted to verify the Cu(100)'s role on C₂H₄ production. While a differential performance was shown, but why the C₂H₄ FE is so much lower? Even one is to ignore the high HER for the moment, a high C₂H₄ selectivity among all carbon-based products should be expected, at least before the single crystal Cu(100) degrades (if it does degrade).

But in the reality even more C1 products (CH₄+ CO) is produced than C2. I feel that this is not only the matter of comparing Cu(100) and Cu(111). Clearly here facet is not the sole control factor (maybe not even the most important one), because if it is, then the single crystal Cu(100) should have the best C₂H₄ selectivity.

Response : Thank you for your comments. Based on the results of the poor selectivity of the C2 product of single crystal Cu(100) foil, the reviewer suggests that the facets are not the only factor influencing the ethylene production and may not even be an important influencing factor. Perhaps the relatively low C2 selectivity of single crystals leads to confusion. It should be emphasized that the introduction of single crystal experiments is only to illustrate the catalytic activity of (100) and (111) crystal planes. Obviously, it is not appropriate to directly compare the selectivity of single crystals with that of nanowire samples, because they exhibit significant differences in both surface and structure. Generally, a catalyst's activity is determined by the intrinsic activity of its surface reaction sites as well as the number of active sites.

(1) In terms of intrinsic activity of crystal surface, we further normalize the partial current density of C₂H₄ products of Cu(100) foil and Cu(111) foil based on ECSA (Fig.R10a) to compare intrinsic activity. The Cu(100) foil shows a superior ECSA-normalized C₂H₄ current density (2.7 mA/mF) over Cu(111) foil (0.8 mA/mF) at -1.4V indicating its better intrinsic activity of C₂H₄. Our nanowire samples, single crystal samples, or current literature reports have confirmed that Cu(100) crystal surface has higher catalytic activity of C₂+ products than

Cu(111) crystal facet [*Chem* 7, 406-420 (2021)] [*J. Mol. Catal. A Chem.* 199,39–47 (2003)] [*ACS Catal.* 7, 1749–1756 (2017)]. But this does not mean that the C₂⁺ selectivity of (100) single crystal foil is definitely high. The same single crystal with (100) faces has different local micro atomic structures, such as step, corner, kink etc., which can lead to significant differences in its catalytic activity. (2) The number of surface sites is also important for the catalytic activity of a catalyst. Taking our CuO-SH catalyst as an example, in terms of ECSA, we found that the ECSA of the CuO-SH electrodes (11.9 mF/cm²) is 18.3 times greater than that of the Cu (100) single crystal foil (0.65 mF/cm²) (Fig.R9). This substantial difference in ECSA between the two electrodes indicates that the number of active sites on Cu foil surfaces is significantly lower than that of CuO-SH. Furthermore, when considering the ECSA-normalized C₂H₄ current density (2.7 mA/mF) of the Cu (100) foil, it closely approximates that of the CuO-SH catalyst (2.9 mA/mF), suggesting a similarity in their intrinsic activity. Therefore, another factor contributing to the performance difference between Cu(100) single crystal foil and CuO-SH (100) rich catalysts could be the difference in the number of surface reaction sites. The above results and analyses fully demonstrate the critical impact of (100) facet on ethylene selectivity. In other words, the key reason for achieving high ethylene selectivity is the improvement of (100) crystal faces.

Our revision: The related modification has been made. Please see the yellow highlighted part on page 28 in the revised supplementary information.

Figure.R10 ECSA-normalized C_2H_4 current density under the different potential of Cu(100) foil and Cu(111) foil, The determination of double layer capacitance(b), and the cyclic voltammety profiles obtained on Cu(100) foil (c) and Cu(111) foil (d) at sweep rates of 20, 40, 60, 80 and 100 $mV s^{-1}$, respectively,

3) I admit that the work reports a high C_2H_4 FE – about the same as the highest previously reported value (in 2020). However, other performance factors are modest: a high over potential results in a modest energy efficiency, and a stability of only 40 hours. To summarize: I am trying to think of a good novelty that can guarantee publication in Nature Communications. At the moment there are two points of potential novelty I can see: 1) a DDT can promote a Cu(100) surface during the electroreduction of CuO, and 2) a ~80% C_2H_4 FE that ties with previous records (at the cost of higher overpotential). I am not sure if the two points of novelty merits publication in Nature Communications. With that said, I am interested to see how the authors may defend the novelty of their

work, I am happy to do another round of review if needed.

Response : Thank you for your comments. In addition to the two innovative points mentioned by the reviewer, we believe that the high selectivity of the single product is also a highlight of this work. Recently, it was widely reported the achievement of total C₂+ products FE of 80-90+% but with a wide product distribution [*J. Am. Chem. Soc.* **145**, 21945–21954 (2023)], [*Angew. Chem. Int. Ed.* e202310788. (2023)]. Due to the complex reaction pathways involved in the reduction of carbon dioxide to multi-carbon products, achieving high selectivity for a single C₂+ product is of great challenge which is critical for industrial production in the e-CO₂RR system. Firstly, the high selectivity could simplify product separation processes, leading to improved separation efficiency and lower production costs. Therefore, improving the selectivity for a single multi-carbon product is of significant importance to the industrial application of electrocatalytic carbon dioxide reduction.

. Moreover, there is also an easily overlooked point. We employed a variety of in-situ techniques to investigate the origins of the enhanced ethylene selectivity and clarified that the high selectivity for ethylene stemmed from the surface modification-driven facet control. The understanding of the catalytic mechanism is of great scientific significance for further improving catalytic activity, selectivity, and stability in the future application of electrocatalytic CO₂RR. In this work, we not only develop a surface modification strategy to improve the single product (C₂H₄) selectivity of CuO nanowire catalysts, but also offer in-depth mechanism analyses, including the critical impact of DDT on the selectivity and stability. In terms of mechanism analysis, although our argument that (100) facet promotes selective improvement has been reported, being recognized and realized are two different things. We have conducted a complete literature review on this matter, as shown in Table R.3. Among this literature, 72% of the papers do not achieve ethylene FE exceeding 60% (48% even did not exceed 50%), and there is no reports of achieving 70% ethylene FE by modulating Cu(100) facets.

Therefore, it remains a substantial challenge through controlling Cu(100) to achieve high ethylene selectivity.

The contributions of this work are summarized as follows. 1) We construct a DDT-treated CuO hierarchical nanoelectrode to achieve high selectivity for ethylene with a Faradaic efficiency of up to 79.5%. 2) Our in-situ investigations and DFT calculations further confirm that the greatly increased selectivity of ethylene is attributed to thiol-stabilized Cu(100). 3) DDT modification is found to not only facilitate CO₂ transfer and enhance *CO coverage on the catalyst surfaces, but also stabilize Cu(100) facet. Inspired by the reviewers' comments, we employed a surface re-treatment to significantly enhance the stability of the CuO-SH catalyst up to 160h. Our findings not only provide an effective strategy to design and construct Cu-based catalysts for highly selective CO₂RR to ethylene, but also offer deep insights into the mechanism of CO₂RR to ethylene. In summary, we believe this work will inspire further effort in this important direction, and it is worthy of publication in Nature Communications.

Table. R3 Sum up of Cu(100)-rich catalysts

	Catalyst	C ₂ H ₄ -FE	reference
1	CuO-C(O)) hybrid	60	Small 2301289 (2023):
2	Ag65–Cu35 JNS-100	54	Adv.Mater. 34 , 2110 (2022)
3	Cu ₂ O-Cu nanocube	56.6	ACS Nano 17 ,12884–12894 (2023)
4	Cu nanocubes	67	J. Mater. Chem. A. 9 , 19932-19939 (2021).
5	Cu nanocube catalyst	60	Nano Lett. 19 , 8461–8468 (2019)
6	Cu nanocubes	33.1	ACS Catal. 2021, 11, 8, 4456–4463
7	Cu meso crystals	27.2	Catal. Sci. Technol. 5 , 161-168 (2015)
8	Cu nanocube catalysts	45	ACS. Nano. 11 , 4825–4831(2017)
9	HRS-Cu	58.6	Nat. Commun. 12, 5745 (2021).
10	NS-D-Cu	32.7	Chinese. J. Catal. 43 , 1066-1073 (2022)
11	Cu-nanocube	32	Nat. Catal. 1 , 111–119 (2018).
12	OD-Cu-III	41.5	J. Am. Chem. Soc. 144 , 259–269 (2022)
13	Cu(OH) 2 -D	58	Angew. Chem. Int. Ed. 60 , 4879–4885 (2021)
14	Cu-CO ₂ -60	68	Nat. Catal. 3 , 98–106 (2020)
15	Au/Cu(100)	25	J. Phys. Chem. C 127 , 3470–3477 (2023)
16	Cu NCs/Al ₂ O ₃ -10C	60.4	Angew. Chem. Int. Ed. 133 , 25042-

			25047.(2021)
17	Cu-IL/GDL	40.67	Fuel 322 , 124103 (2022).
18	wrinkled Cu film	40	ACS. Catal. 11 , 5658–5665 (2021)
19	Cu cubes	57	ACS. Catal. 10 , 4854–486 (2020)
20	ED-Cu(Cl)	58	Chem. Commun. 59 , 10428-10431 (2023)
21	copper hollow fiber	27	Energy. Environ. Sci. 15 , 5391-5404 (2022)
22	Cu nanocrystal cubes	41	Angew. Chem. Int. Ed. 55 , 5789-5792 (2016)
23	(100)-textured Cu foil	67	ACS Appl. Mater. Interfaces 13 , 14050-14055(2021)
24	Cu-NNS	15	Chinese. J. Catal. 43 , 519-525 (2022)
25	CuO-C(O)) hybrid	60	Small. 2301289 (2023).

Reviewer #3 (Remarks to the Author):

In this revised manuscript, the authors made thorough responses to all the comments from the reviewers. Additional experiments were conducted for some of the comments. The experiments were well designed and the obtained results provide solid evidence to answer the questions raised by the reviews and well support the authors' claims. The authors also made corresponding changes and corrections based on the reviewers' comments. The revised manuscript provides a promising route to develop new CO₂RR catalysts toward C₂ products with high selectivities by using thiol (DDT) to stabilize the Cu (100) facet. This work will benefit the understanding of CO₂RR catalysts design toward C₂ products and the mechanism of CO₂RR to C₂ products. The revised manuscript was well organized and written. The experiment results and data analysis support the claims. Therefore, the revised manuscript is recommended for acceptance after addressing the following comments:

1) For the flow cell experiment, were the anolyte and catholyte circulated? In Line 451, it was said "Two parallel fluxes 1 M KOH were injected". Based on Fig. R23, it seems the electrolyte was circulated. Was a reference electrode used?

Response : Thanks for your comments. In the flow cell experiment, as shown in Figure R10, the electrolyte for the cathode and anode is circulated from two lines, green and red, respectively. Moreover, an Hg/HgO electrode was employed as reference electrode.

Figure.R10 The image of the used Flow cell setup.

2)The authors used several pale colors for plots, labels and titles in the figures (yellow, for instance), which is not easy to see for the readers (fig1e, fig 4a, b, fig 5 a, b, c). It will be much better if a dark color is used for those.

Response : Thanks for your comments. To improve the presentation of the data, we have adjusted the colours of the corresponding images.

Our revision: According to the reviewer's comment, the related figure has been added in the revised version. Please see the yellow highlighted part on page 6 (line 141), page 14 (lines 340-341), and page 17 (lines 403-406) in the revised supplementary information.

3)Please check the grammar and format of the manuscript.

Response : Thanks for your comments. According to your suggestions, the whole manuscript has been carefully checked.

Our revision: According to the reviewer's comments, the corresponding revisions have been made. Please see the yellow highlighted part on page 7(line 90), page 14 (line 331), page 15 (line 372) and page 27 (line 645).

REVIEWER COMMENTS

Reviewer #1 (Remarks to the Author):

The authors have done a great job highlighting that their CuO-SH catalyst is indeed different from CuO by addressing the reviewer's comments. While the reactivity data looks good, the spectroscopic data still needs some work before it is ready to publish.

Major Comments

- 1) In supplementary figure 30, having higher peak position wavenumber for Cu(100) than Cu(111) signifies a weaker CO binding to the Cu surface not a stronger one. Similar conclusion for the higher wavenumber CO peak position on CuO-SH compared to CuO. Thus, the authors claims of higher peak position leading to stronger binding for the CuO-SH and Cu (100) are not true.
- 2) If peak position of CO on Cu(100) matches that of CuO-SH, the authors could say that the CuO-SH surface is likely dominated by Cu(100) whereas if CO peak position of CuO matches Cu(111) standard then they can suggest that CuO likely has a Cu(111) like surface but making claims about stronger binding on Cu(100) given the peak positions shown would be incorrect. If making this less bold claim as suggested, then all mention of binding strength would have to be removed from the ATR-IR and DFT calculations.
- 3) In Figure 5., claiming the CO adsorption band on CuO-SH is stronger than the CO adsorption band on CuO is tough to believe if there is no standard peak used as a comparison between the two figures. Electrode thickness or any minor differences in setup with the two catalysts could lead to the differences in signal and lead to different peak intensities.

Minor Comments

- 1) Caption in Supplementary Figure 7 has some repeated text in the last sentence that can be deleted.

Reviewer #2 (Remarks to the Author):

Y. Yao et al. has replied to each and every of my questions with additional data. However, I regret to say that I still cannot recommend publication of the manuscript at its present form. In short, I do not think the novelty of the work has been defended to a satisfactory. I will comment on each of the answers to my questions from the 2nd round.

1. The authors retreated the catalyst using DDT-ethanol every ~50 hours in the time-lapse test. To me this is a perfect experiment to prove the role of the surface thiol group, but if this mandatory surface decoration is itself unstable, then the potential novelty on stability of the work will be lost.

2. My original question was about the competing selectivity (instead of reaction speed or reactivity) of CH₄ and C₂H₄ on single crystal Cu(100) and Cu(111) surfaces. But I can take the authors' reply on normalizing to the ECSA for comparable specific current density for now.

3. The C₂H₄ FE is amongst the highest of existing works, but not the single highest – as listed in Table S2. The authors mentioned potential novelties other than the C₂H₄ FE and the thiol decoration, including in-situ investigations and DFT. If these form the core of the work, the manuscript (including the title) should be arranged around highlighting the innovation in these points (instead of only using them as tools), so that these will not be overlooked.

Point-by-point responses to Reviewers' Comments

Reviewer #1 (Remarks to the Author):

The authors have done a great job highlighting that their CuO-SH catalyst is indeed different from CuO by addressing the reviewer's comments. While the reactivity data looks good, the spectroscopic data still needs some work before it is ready to publish.

Our response: We thank the reviewer for the positive comments, and appreciate the thoughtful and constructive suggestions to improve the overall quality of the work. All these insightful suggestions raised by the reviewer have been thoroughly considered and the corresponding revisions have been made as below.

Major Comments

1) In supplementary figure 30, having higher peak position wavenumber for Cu(100) than Cu(111) signifies a weaker CO binding to the Cu surface not a stronger one. Similar conclusion for the higher wavenumber CO peak position on CuO-SH compared to CuO. Thus, the authors claim of higher peak position leading to stronger binding for the CuO-SH and Cu (100) are not true.

Response: We thank the reviewer for the valuable comments. Indeed, as the reviewer pointed out, it is not rigorous to judge stronger binding for the CuO-SH and Cu (100) based solely on higher peak positions. There are currently two viewpoints in the literature regarding this issue. Most of researchers suggest that a low wavenumber detected CO may bind more strongly to the Cu surface. This is posited on the premise that the robust Cu-carbon bond weakens the carbon-oxygen bond strength, resulting in a lower wavenumber for the C-O stretching mode [*J. Phys. Chem. C.* **122**, 26489–26498 (2018), *J. Phys. Chem.* **68**, 2772– 2777 (1964)]. Conversely, a few hold that a higher-wavenumber adsorbed CO_L peak corresponds to CO_L with a stronger binding strength. This is attributed to CO adsorption occurring on step/defect sites of the Cu surface [*Angew. Chem. Int. Ed.*, **60**, 4879–4885 (2021)]. Generally, it is

suggested that molecules bind more strongly to defect sites than to normal sites due to coordinatively unsaturated structure [*Surf. Sci. Rep.* **6**, 51-94 (1992)]. In our last reply, there was indeed a lack of strong evidence to support it.

To further clarify this issue in detail, inspired by the comments of the reviewers, we analyzed the disparity of the CO adsorption strength between Cu(100) and Cu(111) foils. We introduced tetramethylammonium bromide (TMAB) molecules to the electrolyte as a standard for normalizing the CO_L intensity in the ATR-IR spectrum collected under CO-saturated 0.1M KHCO₃. TMAB was validated to exhibit no noticeable influence on the performance of the CO₂RR, as demonstrated in Fig. R1a, which is consistent with prior results. [*J. Am. Chem. Soc.* **144**, 6613–6622 (2022)]. *In-situ* ATR-FT-IR spectra of TMAB, Cu(100) and Cu(111) foils are given in Fig. R1b-d.

In Fig. R1e, it is obvious that the normalized intensity of *CO_L adsorbed on Cu(100) foil exceeds that on Cu(111) foil at various potentials, indicating higher *CO coverage for Cu(100). These results confirm stronger CO binding at Cu(100) facet. Regarding *In-situ* ATR-FT-IR spectra of CuO-SH and CuO, please refer to the third response. In line with the DFT results, it is affirmed that the Cu(100) facet serves as a more potent *CO adsorption site compared to Cu(111), providing a rationale for the heightened CO_L binding ability of CuO-SH with greater exposure to Cu(100) facet.

According to the reviewer's comments, in order to avoid confusion, we have deleted the discussion on the higher peak position and CO binding strength between CuO-SH and CuO in the Supplementary information.

Figure.R1, **a** Faradaic efficiencies of CO₂ reduction products on the Cu(100) foil as a function of different applied potentials in 0.1M KHCO₃ with/ without 5mM TMAB, **b** the FT-IR spectrum of TMAB, 0.1M HCO₃ and 0.1M KHCO₃ with TMAB, *In-situ* ATR-FT-IR spectra of **c** Cu(100) foil and **d** Cu(111) collected in CO-saturated 0.1M KHCO₃ and **e** the normalized intensity of peaks at 2000-2100 cm⁻¹ that correspond to CO_L intermediates.

Our revision: According to the reviewer's constructive comments, the corresponding revisions have been made. Please see the yellow highlighted

part on page 34 in the revised supplementary information

2) If peak position of CO on Cu(100) matches that of CuO-SH, the authors could say that the CuO-SH surface is likely dominated by Cu(100) whereas if CO peak position of CuO matches Cu(111) standard then they can suggest that CuO likely has a Cu(111) like surface but making claims about stronger binding on Cu(100) given the peak positions shown would be incorrect. If making this less bold claim as suggested, then all mention of binding strength would have to be removed from the ATR-IR and DFT calculations.

Response : We appreciate your valuable comments. In response to the reviewer's comments, we have removed the discussion regarding the relationship between the peak position of adsorbed CO and binding strength to CuO-SH and CuO surface sites. Moreover, the following words has been added into the revised Supplementary information: "The peak position of CO adsorbed on CuO-SH closely overlapped with that on Cu(100) foil, while the CO peak position of CuO is closer to Cu(111) foil. Based on *in situ* ATR-FT-IR spectra through internal standard as shown in Supplementary Fig. 31, furthermore, the normalized intensity of the CO band on Cu(100) is stronger than that on Cu(111). Additionally, the CuO-SH sample exhibits a higher normalized CO intensity compared to the CuO sample, as shown in Supplementary Fig. 30. These results indicate that, compared with CuO catalysts, the greater exposure of CuO-SH catalyst to Cu(100) facets benefits the enhancement of CO binding strength. This is attributed to the stronger CO binding properties of Cu(100) facets."

Our revision: According to the reviewer's constructive comments, the corresponding revisions have been made. Please see the yellow highlighted part on page 15 (lines 361-366) in the revised manuscript and page 35 in the revised supplementary information

3) In Figure 5, claiming the CO adsorption band on CuO-SH is stronger than the CO adsorption band on CuO is tough to believe if there is no standard peak used as a comparison between the two figures. Electrode thickness or any minor differences in setup with the two catalysts could lead to the differences in signal and lead to different peak intensities.

Response : Thanks for your strict comments. We fully agree with the reviewer's comments. In the absence of internal standards, the comparison of peak strengths is very rough. According to the reviewer's comments, we have introduced TMAB as an internal standard (the C-H peak of TMAB at 2920 cm^{-1}) for comparing peak intensities of the CO adsorption band between CuO-SH and CuO (Fig. R3a-b). As illustrated in Fig. R3c, the normalized intensity of $^*\text{CO}$ on the CuO-SH catalyst surpasses that of $^*\text{CO}$ on CuO, indicating a higher $^*\text{CO}$ coverage maintained on the CuO-SH surface. This observation aligns with the results collected without TMAB.

Figure.R1, *In-situ* ATR-FT-IR spectra of **a** CuO-SH sample and **b** CuO sample collected in CO₂-saturated 0.1M KHCO₃ and **c** the normalized intensity of peaks at 2000-2100 cm⁻¹ that correspond to CO_L intermediates.

Our revision: According to the reviewer's constructive comments, the figure and the related discussions have been added in the revised supplementary information. Please see the yellow highlighted part on the page 33 in the revised supplementary information.

Minor comments

1) 1) Caption in Supplementary Figure 7 has some repeated text in the last sentence that can be deleted.

Response : Thank you for carefully reviewing. The repeated texts have been deleted. Moreover, we have read the Supplementary information very carefully again to ensure the accuracy of our paper.

Our revision: The related modification has been made. Please see the yellow highlighted part on page 12 in the revised supplementary information.

Reviewer #2 (Remarks to the Author):

Y. Yao et al. has replied to each and every of my questions with additional data. However, I regret to say that I still cannot recommend publication of the manuscript at its present form. In short, I do not think the novelty of the work has been defended to a satisfactory. I will comment on each of the answers to my questions from the 2nd round.

Our response: We thank the reviewer for the partially positive feedback on this revised manuscript. We have thoroughly considered your comments, especially for the innovation of the article. During the initial review, actually, the reviewer didn't raise any questions regarding the novelty of the work. During the second review, however, the reviewer questioned the innovation of this work due to the stability of only 40 hours. There were two possible reasons: 1) the title and some of the wording in the article misled the reviewer, leading him to deem that stability is one of the focus we emphasized. 2) The stability of 40 hours is indeed unsatisfactory. Although stability still needs to be improved, to be honest, it is not the focus of this work. In this work, we not only develop a surface modification strategy to enhance the single-product (C_2H_4) selectivity and achieve the almost highest FE of existing works, but also provided comprehensive mechanistic understanding through various *in-situ* techniques including *in-situ* XAFS, XRD, IR and Raman and DFT calculation. These analyses elucidate DDT's pivotal influence on selectivity. We believe this work will inspire further effort in this important direction, and shine a light on the underlying mechanism, which is worthy of publication in Nature Communications.

1. The authors retreated the catalyst using DDT-ethanol every ~50 hours in the time-lapse test. To me this is a perfect experiment to prove the role of the surface thiol group, but if this mandatory surface decoration is itself unstable, then the potential novelty on stability of the work will be lost.

Response : We appreciate your fair and objective evaluation. Indeed, the

stability still needs to be improved. But we have to emphasize that it is not the focus of the work. The contributions of the work are summarized as following: 1) we develop a simple surface modification strategy to construct Cu-based catalysts for highly selective CO₂ reduction to ethylene. 2) We achieve high selectivity for ethylene with a faradaic efficiency of up to 79.5%. 3) Based on *in-situ* techniques and DFT calculations, the mechanism of the significantly increased ethylene selectivity is attributed to thiol-improved Cu(100) exposure. Moreover, DDT modification is found to not only facilitate CO₂ transfer and enhance *CO coverage on the catalyst surfaces but also boost the Cu(100) facet. Our findings not only provide an effective strategy for designing and constructing Cu-based catalysts for highly selective CO₂ reduction to ethylene but also offer profound insights into the mechanism of CO₂ reduction to ethylene.

2. My original question was about the competing selectivity (instead of reaction speed or reactivity) of CH₄ and C₂H₄ on single crystal Cu(100) and Cu(111) surfaces. But I can take the authors' reply on normalizing to the ECSA for comparable specific current density for now.

Response : Thank you for your comments. Ethylene production on Cu(100) has been reported as a potential-dependent process, making the ethylene selectivity sensitive to overpotentials [ACS Catal. **10**, 1754–1768 (2020)]. Ethylene could be the major product at lower overpotential, whereas increasing overpotential increases the methane selectivity on Cu(100). To make a more detailed comparison of CH₄ and C₂H₄ selectivity on single crystal Cu(100) and Cu(111) surfaces, we conducted performance tests across a wider range of potential points, from -0.8V to -1.2V in 0.1M KHCO₃. Compared to Cu(111) foil, Cu(100) foil exhibits a maximum FE of 25.4% for ethylene at -1.05V, while that of Cu(111) foil is 5.03%. In addition, Cu(100) foil obviously exhibits a larger ratio of C₂H₄ to CH₄ over a wider range of potential from -0.9V to -1.2V. These results confirm that C₂H₄ generation is more favorable on the Cu(100) facet than on the Cu(111) facet. As a modeling experiment to validate C₂H₄ selectivity

between Cu(100) and Cu(111) facets, the results of the single crystal Cu foil experiment support our argument concerning the higher exposure of Cu(100) facets in CuO-SH, resulting in superior C₂H₄ selectivity.

Figure.R3, **a** Faradaic efficiencies of CO₂ reduction products as a function of a function of applied potential over Cu(100) and Cu(111) measured in 0.1M KHCO₃ and **c** the ratio of C₂H₄ to CH₄.

Our revision: The related modifications have been made. Please see the yellow highlighted part on page 27 in the revised supplementary information.

3. The C₂H₄ FE is amongst the highest of existing works, but not the single highest – as listed in Table S2. The authors mentioned potential novelties other than the C₂H₄ FE and the thiol decoration, including in-situ investigations and DFT. If these form the core of the work, the manuscript (including the title) should be arranged around highlighting the innovation in these points (instead of only using them as tools), so that these will not be overlooked.

Response : Thanks for your valuable comments. We emphasize that the contributions of this work include the following aspects: 1) we develop a facile and efficient surface strategy to significantly improve the C₂H₄ selectivity. 2) The C₂H₄ FE is pretty high in existing works. 3) The underlying mechanisms why the C₂H₄ selectivity significantly improve are clarified by various *in-situ* techniques. According to the reviewer's suggestions, we have revised the manuscript to highlight the novelty of our findings, including the title was revised to “A surface strategy to boost the selectivity and *in-situ* mechanism insight into electrocatalytic CO₂ reduction to ethylene”.

Our revision: According to the reviewer's constructive comments, the related modifications have been made. Please see the yellow highlighted part on page 1 (lines 1-3, lines 28-29) in the revised manuscript.

REVIEWERS' COMMENTS

Reviewer #1 (Remarks to the Author):

I am mostly satisfied with the authors responses to my questions. I have only one minor comment:-

1) In the main paper on line 361-366, the authors mention that CuO-SH has higher CO binding peak intensity than CuO and it is likely due to the presence of more Cu(100) sites. While they explain this logic well of how CuO-SH relates to having more Cu(100) in the SI, in the main paper the train of thought doesn't seem logical. The authors first mention that the CuO-SH has a stronger linearly bonded CO band than CuO. Next, they mention that the stronger coverage can be attributed to Cu(100) without a reference. I feel these two points are missing the connecting statement that linearly bonded CO on CuO-SH peak position matches Cu(100) while CuO peak position matches Cu(111) which is only mentioned in the SI. I encourage the authors to move that statement to the main paper too so it's easier for the authors to follow their train of thought.

Point-by-point responses to Reviewers' Comments

Reviewer #1 (Remarks to the Author):

I am mostly satisfied with the authors responses to my questions. I have only one minor comment:

Our response: We thank the reviewer for the positive comments, and appreciate the thoughtful and constructive suggestions to improve the overall quality of the work. All these insightful suggestions raised by the reviewer have been thoroughly considered and the corresponding revisions have been made as below.

1) In the main paper on line 361-366, the authors mention that CuO-SH has higher CO binding peak intensity than CuO and it is likely due to the presence of more Cu(100) sites. While they explain this logic well of how CuO-SH relates to having more Cu(100) in the SI, in the main paper the train of thought doesn't seem logical. The authors first mention that the CuO-SH has a stronger linearly bonded CO band than CuO. Next, they mention that the stronger coverage can be attributed to Cu(100) without a reference. I feel these two points are missing the connecting statement that linearly bonded CO on CuO-SH peak position matches Cu(100) while CuO peak position matches Cu(111) which is only mentioned in the SI. I encourage the authors to move that statement to the main paper too so it's easier for the authors to follow their train of thought.

Response: We thank the reviewer for the valuable comments. To strengthen the manuscript's coherence, a connecting statement has been incorporated in the revised version. This statement highlights that the peak position of linearly bonded CO on CuO-SH corresponds to Cu(100), whereas the CuO peak position aligns with Cu(111).

Our revision: According to the reviewer's constructive comments, the corresponding revisions have been made. Please see the yellow highlighted part in line 337-342 on page 12 in the revised manuscript.